# Train on Pins and Test on Obstacles for Rectilinear Steiner Minimum Tree

**Xingbo Du, Ruizhe Zhong, Junchi Yan**[*]
School of Computer Science & School of Artificial Intelligence, Shanghai Jiao Tong University
{duxingbo,zerzerzerz271828,yanjunchi}@sjtu.edu.cn

## Abstract

Rectilinear Steiner Minimum Tree (RSMT) is widely used in Very Large Scale Integration (VLSI) and aims at connecting a set of pins using rectilinear edges while minimizing wirelength. Recently, learning-based methods have been explored to tackle this problem effectively. However, existing methods either suffer from excessive exploration of the search space or rely on heuristic combinations that compromise effectiveness and efficiency, and this limitation becomes notably exacerbated when extended to the obstacle-avoiding RSMT (OARSMT). To address this, we propose OAREST, a reinforcement learning-based framework for constructing an Obstacle-Avoiding Rectilinear Edge Sequence (RES) Tree. We theoretically establish the optimality of RES in obstacle-avoiding scenarios, which forms the foundation of our approach. Leveraging this theoretical insight, we introduce a dynamic masking strategy that supports parallel training across varying numbers of pins and extends to obstacles during inference. Empirical evaluations on both synthetic and real-world benchmarks show superior effectiveness and efficiency for RSMT and OARSMT problems, particularly in handling obstacles without training on them. Code available: https://github.com/Thinklab-SJTU/EDA-AI/.

## 1 Introduction

Rectilinear Steiner Minimum Tree (RSMT) and its obstacle-avoiding version named OARSMT are critical combinatorial problems in electronic design automation (EDA) in Very Large Scale Integration (VLSI) design [1, 2, 3]. As an NP-complete problem [4], the goal of RSMT is to connect a given set of pins using rectilinear edges, i.e., edges that are parallel to the axes, while minimizing the total wirelength. Recently, the RSMT problem [5, 6, 7, 8, 9], as well as the entire EDA community [10, 11, 12, 13], has received extensive attention from the machine learning (ML) community. However, with the increasing number of nets in modern technologies and particularly the incorporation of obstacles, existing ML-based methods suffer from various challenges.

We summarize the critical **challenges** for ML-based approaches in both vanilla RSMT and OARSMT problems in Table 1, including: 1) *Lack of efficient and accurate representations*: Efficient representations are essential for reducing the searching space (e.g., the action space in the reinforcement learning (RL) agent), which can otherwise become excessively complex and computationally exhaustive. Accurate representations, on the other hand, enable ML approaches to learn the global optimum in an end-to-end manner without relying on post-processing [7, 9] or integrating heuristics [8]. 2) *Multi-degree GPU parallelization*: Existing ML-based techniques [5, 6] typically require inputs of the same size, limiting their ability to handle multi-degree instances (i.e., instances with varying numbers of pins) in parallel. 3) *Obstacle avoidance*: The introduction of obstacles significantly increases problem complexity and further exacerbates the above two challenges.

---

[*]Correspondence to: Junchi Yan. This work was partly supported by NSFC (92370201).

Table 1: Comparisons of ML-based methods.

| Method | Representation | End-to-End | Multi-Degree GPU-Parallelization | Obstacle Avoidance |
|--------|----------------|------------|----------------------------------|--------------------|
| REST [5] | RES (Optimal) | ✓ | ✗ | ✗ |
| EPST [6] | EPS | ✓ | ✗ | ✗ |
| Chen et al. [7] | Steiner Points | ✗ | ✓ | ✓ |
| Lin et al. [8] | OASG + SPF | ✗ | ✓ | ✓ |
| Chen et al. [9] | Steiner Points | ✗ | ✓ | ✓ |
| OAREST (ours) | RES (Optimal) | ✓ | ✓ | ✓ |

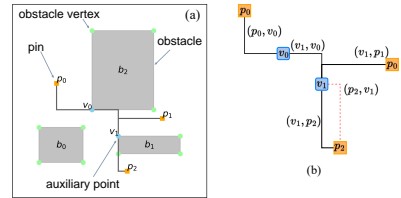

Figure 1: a) Different components. b) Rectilinear edges in an OARSMT.

To address these challenges, we first extend the rectilinear edge sequence (RES) representation, originally proposed in REST [5], to the OARSMT problem. Such an extension is nontrivial as the problem's structure is fundamentally altered. We theoretically demonstrate that RES can achieve optimality for both RSMT and OARSMT problems.

Building upon these theoretical insights, we propose a novel RL-based framework OAREST (Obstacle-Avoiding Rectilinear Edge Sequence Tree), designed to tackle multi-degree GPU parallelization and dynamic obstacle avoidance simultaneously. Within this framework, we introduce a dynamic masking strategy that selectively activates candidate corner vertices of obstacles, ensuring efficient and obstacle-aware decision-making. Empirical results on both synthetic and industrial benchmarks validate the effectiveness and efficiency of OAREST in solving RSMT and OARSMT problems. Particularly, the obstacles can be avoided effectively in OARSMT problems without training on any obstacles. **The main contributions are listed as follows:**

1) **Theoretical findings and derived RL framework**: We prove the optimality of the rectilinear edge sequence (RES) representation for the obstacle-avoiding rectilinear Steiner minimum tree (OARSMT) problem. Inspired by this, we introduce OAREST, an end-to-end RL framework based on the RES, which combines theoretical optimality with computational efficiency. To the best of our knowledge, OAREST is the first ML-based approach handling obstacles without training on obstacles.

2) **Novel dynamic masking strategy**: Within the RL framework, we devise a novel dynamic masking strategy that can not only handle multi-degree parallel training/inference on GPUs but also dynamically activate candidate obstacle vertices for effective obstacle avoidance. Thus, the OARSMT problems can be handled without training on any obstacles.

3) **Empirical advancements and efficient obstacle handling**: We perform experiments on extensive instances with/without obstacles and demonstrate comparable performance to advanced baselines. The results particularly highlight OAREST's robust capabilities in managing parallel inference and avoiding obstacles efficiently.

## 2 Preliminaries and Main Theorem

Before introducing the RL framework, we first propose key definitions in Sec. 2.1 and state the OARSMT problems in Sec. 2.2, based on which we introduce the main theorem in Sec. 2.3 that demonstrates the optimality of rectilinear edge sequence (RES) in OARSMT problems.

### 2.1 Definition

We introduce various definitions in OARSMT problems and visualize components in Fig. 1(a).

**Definition 2.1. [Rectilinear Steiner Tree, RST]** Given a finite set of pins[2] $\mathcal{P} = \{p_0, p_1, \cdots, p_{n-1}\}$ with each pin $p_i$ having fixed coordinates $(x_i, y_i)$, a rectilinear Steiner tree (RST) is a tree that connects all pins and some non-pin auxiliary points, where each edge is parallel to the coordinate axis. The non-pin auxiliary points are named **Steiner points**, which are used to minimize the total length.

Here, each edge is either horizontal or vertical. In OARSMT problems, RSTs are required to avoid overlapping with rectangular obstacles, which are defined as:

---

[2]We use 'pin' to distinguish from 'obstacle vertex' and 'Steiner points', and collectively name them 'points'.

**Definition 2.2. [Rectangular Obstacles]** Define $\mathcal{O} = \{o_0, o_1, \cdots, o_{m-1}\}$ as a group of rectangular regions. Each obstacle $o_i = (v_i^{(ld)}, v_i^{(lu)}, v_i^{(ru)}, v_i^{(rd)})$ is composed of four vertices in its left-down (ld), left-up (lu), right-up (ru), and right-down (rd) corners. Denote these vertices as $\mathcal{V}(\mathcal{O})$.

Note that any complex rectilinear obstacle (e.g., rectilinear polygons) can be easily achieved by combining different rectangular obstacles. To formulate RSTs in an OARSMT problem, we apply the rectilinear edge sequence (RES) [5] representation to our framework, defined as:

**Definition 2.3. [Rectilinear Edge Sequence, RES]** For a given set of pins $\mathcal{P} = \{p_0, p_1, \cdots, p_{n-1}\}$, its RES is a sequence with $n-1$ rectilinear edge pairs $((v_0, h_0), (v_1, h_1), \cdots, (v_{n-2}, h_{n-2}))$, where $v_i, h_i \in \{0, 1, \cdots, n-2\}$ denotes the indices of pins.

Note that rectilinear edges are non-commutative in rectilinear geometry, i.e., $(v, h) \neq (h, v)$. Specifically, $(v, h)$ denotes a path where the vertical edge starts at $v$ followed by a horizontal edge ending at $h$. A visualized example of rectilinear edges is shown in Fig. 1(b), where $(p_2, v_1)$ and $(v_1, p_2)$ are two distinct rectilinear edges. To keep RES always valid in RSMT construction problem, RES is required to satisfy: 1) $v_0 \neq h_0$; 2) For each rectilinear edge pair $(v_i, h_i)$, exactly one of $v_i, h_i$ is visited before by previous rectilinear edge pair, and the other is not. Details given by [5] is moved to Lemma A.4 in Appendix A. Other definitions that are only related to the proof of our main theory, including Hanan Grid and extended Hanan Grid, can be found in Appendix A.1.

## 2.2 Problem Statement

Based on the definitions, we formulate OARSMT as:

**Definition 2.4. [Obstacle-Avoiding Rectilinear Steiner Tree, OARSMT]** Given a finite set of pins $\mathcal{P} = \{p_0, p_1, \cdots, p_{n-1}\}$ and a set of rectangular obstacles $\mathcal{O} = \{o_0, o_1, \cdots, o_{m-1}\}$, an obstacle-avoiding rectilinear Steiner tree (OARSMT) is defined as:

$$T^* = \underset{T \in \mathcal{T}}{\operatorname{argmin}} \sum_{e \in \mathcal{E}(T)} |e|, \tag{1}$$

under constraints: 1) $T$ is a tree connecting all pins in $\mathcal{P}$, 2) $\forall e \in \mathcal{E}(T)$: $e$ is parallel to coordinate axes, 3) $\forall e \in \mathcal{E}(T), \forall o_i \in \mathcal{O}$ : $e \cap \operatorname{int}(o_i) = \emptyset$, 4) $\mathcal{V}(T) \setminus \mathcal{P}$ are Steiner points. Here, $\mathcal{T}$ is the set of all rectilinear Steiner trees in terms of $\mathcal{P}$ and $\mathcal{O}$. Additionally, $|e|$ denotes the Manhattan length of edge $e$, $\mathcal{E}(T)$ represents the set of edges in tree $T$, $\mathcal{V}(T)$ is the set of vertices in $T$, and $\operatorname{int}(o_i)$ denotes the interior region of obstacle $o_i$.

Intuitively, OARSMT aims to obtain a rectilinear tree with minimum length that avoids obstacles.

## 2.3 Main Theorem

**Theorem 2.5. [Optimality of RES in OARSMT]** *For any set of pins $\mathcal{P} = \{p_0, p_1, \cdots, p_{n-1}\}$ and a set of rectangular obstacles $\mathcal{O} = \{o_0, o_1, \cdots, o_{m-1}\}$, an optimal RES of $\mathcal{P} \cup \mathcal{V}'(\mathcal{O})$ can always be found such that its corresponding tree is an optimal OARSMT for $\mathcal{P}$ under obstacles $\mathcal{O}$. Here, $\mathcal{V}'(\mathcal{O}) \subset \mathcal{V}(\mathcal{O})$ is a subset of the corner vertices of all obstacles.*

This theorem demonstrates that an optimal RES can always be found to construct an optimal OARSMT. The proof can be found in Appendix A.3. Note that Theorem 2.5 gives us an insightful **intuition**: *In order to construct an OARSMT, we are only required to construct a RES that connects all pins and a subset of corner vertices of obstacles.*

## 3 Obstacle-Avoiding Rectilinear Edge Sequence Tree

**Overview.** We propose an RL-based framework to handle OARSMT. The overall pipeline is given in Fig. 2, while the connecting process of the rectilinear edge sequence (RES) is shown by a simple toy example in Fig. 2(d). Following [5], we use actor-critic networks to generate rectilinear edge sequence (RES) in Sec. 3.1. To accommodate multi-degree parallelization and obstacle handling, we introduce a novel dynamic masking strategy in Sec. 3.2. In Sec. 3.3, we provide a detailed discussion on our focus on multiple small instances in OARSMT problems and analyze the time complexity of the proposed framework.

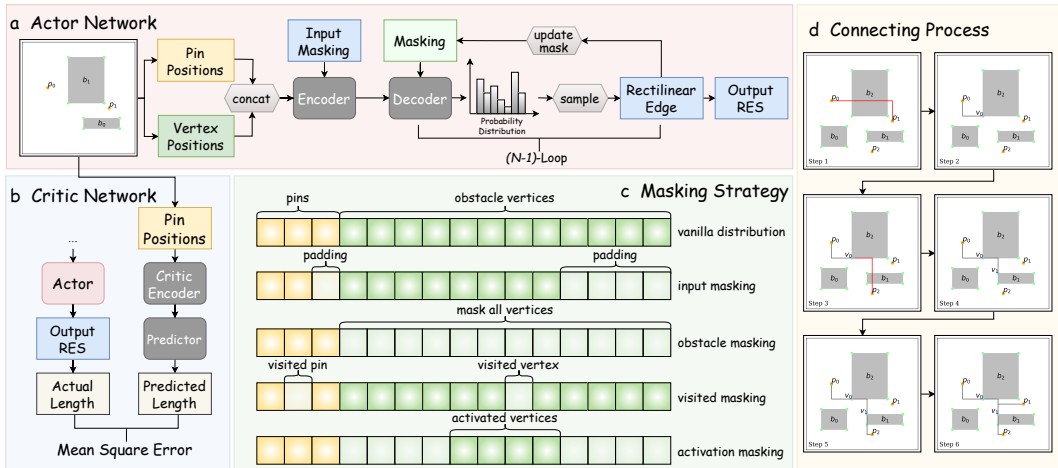

Figure 2: Pipeline of OAREST. a) Actor network that generates the rectilinear edge sequence (RES) step-by-step with dynamic masking strategies. b) Critic network that fits the actual length of RES. c) Dynamic masking strategies, including input masking, obstacle masking, visited masking, and activation masking, that enable the RES generation with multi-degree forward passing and obstacle avoidance. d) Visualization of the connecting process of OAREST on a toy sample with 3 pins $(p_0, p_1, p_2)$ and 3 obstacles $(b_0, b_1, b_2)$.

## 3.1 Generate RES using Actor-Critic Network

Similar to the objectives of REST [5], we utilize an actor-critic network to learn the construction of the rectilinear edge sequence (RES). Given the positions of all pins $\mathcal{P} = \{p_0, p_1, \cdots, p_{n-1}\}$, the actor network $\mathrm{Actor}(r|\mathcal{P}; \xi)$ with parameters $\xi$, as shown in Fig. 2(a), takes a series of sequential actions to generate the elements of the RES $r \in \mathcal{R}(\mathcal{P})$ and yields the probability of generating $r$, where $\mathcal{R}(\mathcal{P})$ is the set of all possible RES for $\mathcal{P}$. The actor network here is trained to generate the optimal RES. The critic network $\mathrm{Critic}(\mathcal{P}; \zeta)$ with parameters $\zeta$, as depicted in Fig. 2(b), predicts the length of the RSMT of $\mathcal{P}$, which is trained to approximate its actual length $L(\mathcal{P}, r)$. Note that $L(\mathcal{P}, r)$ is actually the evaluation of the RES, which can be achieved linearly [5] and is assumed to approximate the actual RSMT length accurately.

More specifically, for each given pin set $\mathcal{P}$, the objective of the actor network is to minimize the expected advantage of the RSMT generated by $\mathrm{Actor}(r|\mathcal{P}; \xi)$, and we use the policy gradient algorithm REINFORCE [14] to compute the gradient:

$$\min_{\xi} \quad \mathbb{E}_{r \sim R(\mathcal{P})} \left(\mathrm{Critic}(\mathcal{P}; \zeta) - L(\mathcal{P}, r)\right) \mathrm{Actor}(r|\mathcal{P}; \xi). \tag{2}$$

Conversely, the critic network's objective is to minimize the mean square error (MSE) between its predicted length and the actual length of RSMT:

$$\min_{\zeta} \quad \mathbb{E}_{r \sim R(\mathcal{P})} \|\mathrm{Critic}(\mathcal{P}; \zeta) - L(\mathcal{P}, r)\|_2^2. \tag{3}$$

In practice, given a batch of pin sets $\{\mathcal{P}_0, \mathcal{P}_1, \cdots, \mathcal{P}_{b-1}\}$, one RES $r_i$ is generated for each $\mathcal{P}_i (i \in \{0, \cdots, b-1\})$. Consequently, Eq. 2 and Eq. 3 can be reformulated as:

$$\min_{\xi} \frac{1}{b} \sum_{i=1}^{b} \left(\mathrm{Critic}(\mathcal{P}_i; \zeta) - L(\mathcal{P}_i, r_i)\right) \mathrm{Actor}(r_i|\mathcal{P}_i; \xi),$$
$$\min_{\zeta} \frac{1}{b} \sum_{i=1}^{b} \|\mathrm{Critic}(\mathcal{P}_i; \zeta) - L(\mathcal{P}_i, r_i)\|_2^2. \tag{4}$$

These objectives are optimized simultaneously. In inference, only $\mathrm{Actor}(r_i|\mathcal{P}_i; \xi)$ is required to generate the RES $r_i$ with the highest probability. Note that both training and inference are end-to-end.

## 3.2 Dynamic Masking

The key issues in Sec. 3.1 are twofold: 1) the actor-critic network requires inputs and outputs of fixed sizes to enable GPU parallelization, and 2) obstacle constraints are not incorporated. To address these issues, as illustrated in Fig. 2(c), we propose a dynamic masking mechanism, which dynamically adjusts valid action spaces during each decision step, enabling efficient handling of variable input sizes and the integration of obstacle constraints.

First, we use a tensor $\mathbf{I} \in \mathbb{R}^{b \times (\hat{n}+4\hat{m}) \times 2}$ to represent the input of the actor net, where $b$ is the batch size, $\hat{n}$ is the max number of pins of the pin set in a batch $\{\mathcal{P}_i\}_{i=0}^{n-1}$, and $\hat{m}$ is the max number of obstacles in $\{\mathcal{O}_i\}_{i=0}^{m-1}$. The value '4' here means the four corner vertices of an obstacle, and '2' means the x- and y-axis of coordinates. For the instances with $n(< \hat{n})$ pins or $m(< \hat{m})$ obstacles, we pad the corresponding elements to -1 in the input tensor. The dynamic mechanism is then devised with the following components:

**Input/Obstacle Masking.** The input mask is devised to ignore the computations for invalid elements. It is constructed using a matrix $\mathbf{M}^{\text{input}} \in \{0,1\}^{b \times (\hat{n}+4\hat{m})}$. The obstacle mask ensures that the actor network does not select an obstacle vertex as the first element of the RES, which is constructed using a matrix $\mathbf{M}^{\text{ob}} \in \{0,1\}^{b \times (\hat{n}+4\hat{m})}$. Each element of $\mathbf{M}^{\text{input}}$ and $\mathbf{M}^{\text{ob}}$ is defined as:

$$\mathbf{M}_{i,j}^{\text{input}} = \begin{cases} 0, & \text{if } \mathbf{I}_{i,j,0} = \mathbf{I}_{i,j,1} = -1, \\ 1, & \text{otherwise}, \end{cases} \qquad \mathbf{M}_{i,j}^{\text{ob}} = \begin{cases} 0, & \text{if } j > \hat{n}, \\ 1, & \text{otherwise}. \end{cases} \tag{5}$$

**Visited/Activation Masking.** Based on the properties of the RES (as described in Sec. 2.1), each rectilinear edge must have exactly one visited and one unvisited element. The visited mask enforces this condition. It is represented by a matrix $\mathbf{M}^{\text{visited}} \in \{0,1\}^{b \times (\hat{n}+4\hat{m})}$. On the other hand, the activation masking activates parts of the obstacle vertices and is used to avoid obstacles when generating the RES, which is initialized as $\mathbf{M}^{\text{act}} = \mathbf{M}^{\text{ob}}$. Each time a newly generated rectilinear edge overlaps with one or more obstacles, the elements in $\mathbf{M}^{\text{act}}$ corresponding to the corner vertices of these obstacles are activated. Each element of $\mathbf{M}^{\text{visited}}$ and $\mathbf{M}^{\text{act}}$ is defined as:

$$\mathbf{M}_{i,j}^{\text{visited}} = \begin{cases} 0, & \text{if } j \text{ is an element in RES } r_i, \\ 1, & \text{otherwise}. \end{cases} \qquad \mathbf{M}_{i,j}^{\text{act}} = \begin{cases} 0, & \text{if } j \text{ is inactivated}, \\ 1, & \text{otherwise}. \end{cases} \tag{6}$$

When a newly generated rectilinear edge does not overlap with obstacles, the mask is re-initialized as $\mathbf{M}^{\text{act}} = \mathbf{M}^{\text{ob}}$. In practice, the actor net sequentially determines which pin or vertex to generate. Each action is based on a probability distribution matrix $\mathbf{S} \in \mathbb{R}^{b \times (\hat{n}+4\hat{m})}$. The combination of dynamic masks is then used to select the elements in RES:

- To begin with, we sample a batch of *start* points from $\mathbf{S}^{\text{start}} = \mathbf{S} \circ (\mathbf{M}^{\text{input}} \wedge \mathbf{M}^{\text{ob}})$, where $\circ$ is the Hadamard product (element-wise multiplication) and $\wedge$ represents logical 'AND', which evaluates to true only when both operands are true. This ensures that only valid pins are selected as the first element of a RES.
- For a batch of rectilinear edges, we select a batch of *visited* points from $\mathbf{S}^{\text{visited}} = \mathbf{S} \circ (\mathbf{1} - \mathbf{M}^{\text{visited}})$. For the same batch of rectilinear edges, we select a batch of *unvisited* points from $\mathbf{S}^{\text{unvisited}} = \mathbf{S} \circ (\mathbf{M}^{\text{input}} \wedge \mathbf{M}^{\text{visited}} \wedge \mathbf{M}^{\text{act}})$. This implies that the selected points must be valid unvisited ones while $\mathbf{M}^{\text{act}}$ ensures that the rectilinear edge does not overlap with any obstacles, as governed by Eq. 6.

Equipped with the dynamic masking strategy, OAREST is capable of addressing the multi-degree training and inference for RSMT problems, and can infer directly on OARSMT problems without training with any obstacle. Functions and executions of different masks are summarized in Table 2.

## 3.3 Remarks

**Intuition on generalization to obstacles and strategy.** Theorem 2.5 suggests that it suffices to construct a RES that connects all pins and a subset of obstacle corner vertices. The RL agent's goal is then to connect these pins and selected corners. Since OAREST is trained only on pins but must also connect corner vertices, a natural idea is to treat corner vertices as pins. However, this raises

Table 2: Functions and executions of different masks.

| Masking | Function | Execution |
|---|---|---|
| **Input** | Achieve multi-degree GPU parallelization. | Mask invalid elements at the beginning. |
| **Obstacle** | 1) Avoid choosing a corner vertex as the first element of the RES; 2) Avoid unnecessary corner vertex connection. | 1) Mask all obstacle vertices at the beginning. 2) When a newly generated rectilinear edge does not overlap with obstacles, re-initialize the activation mask as the obstacle mask. |
| **Visited** | Construct valid RES following the criterion of Lemma A.4. | Use masking to generate a visited point and an unvisited point for each rectilinear edge. |
| **Activation** | Avoid generating rectilinear edges that overlap with obstacles. | Each time a newly generated rectilinear edge overlaps with one or more obstacles, the elements in the activation masking corresponding to the corner vertices of these obstacles are activated. |

two issues: (1) not all corner vertices need to be connected, and (2) the resulting edges may overlap obstacles. Accordingly, obstacle masking is used to avoid unnecessary corner-vertex connections (issue 1), and an activation mask is used to prevent overlap with obstacles (issue 2). Notably, both masking strategies can be applied purely at inference.

**Time Complexity for Inference.** The overall time complexity for inference is $O\left((\hat{n} + \hat{m}) \cdot T_{\text{batch}}/b\right)$, where $T_{\text{batch}}$ represents the inference time for a single batch, and $b$ denotes the batch size. This complexity arises because all computations within the actor network are linear with respect to $\hat{n} + \hat{m}$. The high efficiency of the framework can thus be attributed to the ability to handle large batch sizes, particularly for multiple small-sized instances.

**Single Large Instance vs. Multiple Small Instances.** Lots of previous works [7, 8, 9] primarily focus on constructing OARSMTs of large-scale instances with hundreds or even thousands of pins; however, these approaches often rely on heuristic tools, which limit their performance in GPU parallelization. In contrast, this paper takes an orthogonal approach by focusing on OARSMT construction for multiple small instances, achieving significant improvements in parallel efficiency. Furthermore, its focus on small-scale instance construction makes it more aligned with practical scenarios in certain industrial applications. As shown in Table 3, we analyze benchmarks from the ICCAD19 [15] global routing contest, a highly regarded industry competition. The results reveal that most nets contain fewer than 10 pins. Specifically, the proportion of nets with more than 100 pins is only 0.002%, highlighting the importance of efficient inference for multiple small instances.

## 4 Experiments

Experiments were conducted on a machine equipped with an AMD EPYC 7402 24-Core Processor, an NV GeForce RTX 4090, and 512 GB of RAM. To ensure robust evaluations, experiments were repeated using three distinct seeds. The experimental protocols are detailed in Section 4.1, while the main results for RSMT/OARSMT are presented in Sections 4.2/4.3. In Sec. 4.4, we perform ablation studies, show generality on untrained degrees, and evaluate the performance on real-world circuits.

### 4.1 Experimental Protocols

**Datasets.** We evaluated the proposed method using two datasets. The first consists of randomly generated test data as described by REST [5]. This dataset includes varying degrees ranging from 5 to 50 in increments of 5, referred to as R5, R10, etc. Each subset contains $10\,\text{k}$ test instances. To assess the method's capability in handling obstacles, we further introduced 5 and 10 random obstacles to the test cases in R5, R10, etc. The second dataset consists of real-world global routing benchmarks from ICCAD19 [15], which is directly evaluated using the model trained on R5–R50.

**Metrics.** We employ the total wirelength as the primary metric for RSMT and OARSMT, which is calculated using the total Manhattan distance of a tree:

$$L = \sum_{e \in \mathcal{E}} \text{length}(e) = \sum_{(v_i, v_j) \in \mathcal{E}} (|x_i - x_j| + |y_i - y_j|), \tag{7}$$

Table 3: Statistics of nets ($\times 10^3$) with different numbers of pins in ICCAD19 [15] benchmark.

| # Pins | ispd18_test{1-10} | | ispd19_test{1-10} | |
|---|---|---|---|---|
| | # Nets ($\times 10^3$) | proportion (%) | # Nets ($\times 10^3$) | proportion (%) |
| <3 | 579 | 57.07 | 1,791 | 57.17 |
| 3-10 | 387 | 38.12 | 1,197 | 38.21 |
| 11-20 | 13 | 1.29 | 40 | 1.28 |
| 21-30 | 4 | 0.42 | 13 | 0.41 |
| 31-40 | 16 | 1.58 | 47 | 1.49 |
| 41-50 | 4 | 0.44 | 13 | 0.42 |
| 51-100 | 11 | 1.08 | 32 | 1.02 |
| >100 | 0.02 | 0.002 | 0.06 | 0.002 |

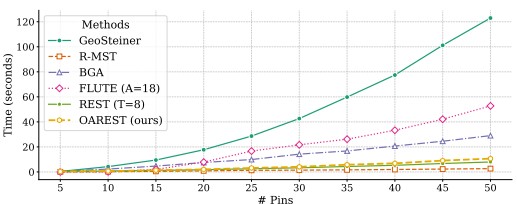

Figure 3: Runtime (seconds) of RSMT methods for testing $10\,\mathrm{k}$ instances on varied pins.

where $\mathcal{E}$ is the edge set of the tree. The average percentage error $(L - L_{\mathrm{opt}})/L_{\mathrm{opt}}$ is then utilized to measure the gap from the optimal solution $L_{\mathrm{opt}}$ obtained by the exact algorithm GeoSteiner [16]. For OARSMT problems, we further employ the overlap metric:

$$\mathrm{overlap}(e, o_k) = \begin{cases} 1, & \text{if } e \cap \mathrm{int}(o_k) \neq \emptyset, \\ 0, & \text{Otherwise,} \end{cases}, \qquad O = \sum_{e \in \mathcal{E}} \sum_{k=1}^{n} \mathrm{overlap}(e, o_k), \qquad (8)$$

where $\mathrm{int}(o_k)$ is the interior region of obstacle $o_k \in \mathcal{O}$.

**Baselines.** RSMT baselines include GeoSteiner [16], R-MST [17], BGA [18], FLUTE [19], and REST [5]. For OARSMT, we evaluate wirelength, overlaps, and runtime against GeoSteiner [16], OARST [20], and OARSMT [21], respectively. Details are shown in Appendix B.2.

**Other Settings and Results.** We show other experimental settings, including the model structures of the actor-critic networks, hyperparameters, training/inference strategies, and more visualizations in Appendix B.3 - B.6.

## 4.2 Main Results of RSMT

Table 4 shows the average percentage error of algorithms compared to the exact algorithm GeoSteiner [16] across various scales of random instances. R-MST [17] has a substantial gap from the optimal GeoSteiner. Other approaches, including BGA [18], FLUTE [19], and REST [5] demonstrate much better performance, achieving an average error of less than $1\%$. In particular, FLUTE (A = 18) and REST (T = 8) perform strongly in all these instances. Building on this strong foundation, OAREST surpasses other baselines on R5–R40 instances and achieves competitive performance on R45 and R50. A visualized example of a 50-pin instance is shown in Fig. 4(a).

The advantages of OAREST are even more pronounced in terms of runtime. In line with the setup in [5], we adopt a batch size of $100\,\mathrm{k}/degree$ for inference. As illustrated in Fig. 3, while GeoSteiner produces optimal results, its runtime grows exponentially with the number of pins. FLUTE (A = 18) offers significantly better efficiency than GeoSteiner, but its runtime still increases considerably as the number of pins grows. In contrast, the inference time of OAREST scales almost linearly with the number of pins, slightly exceeding that of REST. The additional time mainly stems from the inspection of obstacles. Note that while REST performs pretty well in both wirelength and runtime, it struggles with the multi-degree inference problem. This limitation results in a sharp increase in runtime when dealing with variable pin counts in real-world circuits. We will further investigate this issue in Sec. 4.4.

## 4.3 Main Results of OARSMT

We evaluate the wirelength, overlap, and runtime of GeoSteiner, OARST, ObSteiner, and OAREST in Fig. 5, where GeoSteiner [16] and ObSteiner [21] are used as the exact solvers for the RSMT and OARSMT problems, respectively. OARST implemented by [20] serves as an approximation solution for OARSMT. In this setting, we randomly add 5 and 10 obstacles to R5-R50 test cases proposed by [5], using a full batch size of $10\,\mathrm{k}$. Fig. 4(b) visualizes an instance with 20 pins and 10 obstacles.

In Fig. 5(a), all algorithms, including the exact OARSMT solver, have a gap compared to GeoSteiner due to the additional complexity of accounting for overlaps. While the wirelength achieved by OAREST is slightly longer than that of the exact OARSMT solution, it demonstrates a clear advantage over the approximation-based OARST.

Table 4: Average percentage error (%) of RSMTs compared to the exact algorithm GeoSteiner [16]. Best results except GeoSteiner are in **bold**.

| Instances | GeoSteiner [16] | R-MST [17] | BGA [18] | FLUTE [19] | | REST [5] | | OAREST (ours) |
|---|---|---|---|---|---|---|---|---|
| | | | | A = 3* | A = 18* | T = 1* | T = 8* | |
| R5 | 0.00 | 10.91 | 0.23 | **0.00** | **0.00** | 0.02 | **0.00** | **0.00 ± 0.000** |
| R10 | 0.00 | 11.96 | 0.48 | 0.12 | 0.04 | 0.23 | **0.01** | **0.01 ± 0.000** |
| R15 | 0.00 | 12.19 | 0.53 | 0.55 | 0.06 | 0.45 | **0.03** | **0.03 ± 0.001** |
| R20 | 0.00 | 12.41 | 0.57 | 1.03 | 0.11 | 0.56 | 0.07 | **0.06 ± 0.001** |
| R25 | 0.00 | 12.47 | 0.58 | 1.44 | 0.18 | 0.69 | 0.12 | **0.10 ± 0.001** |
| R30 | 0.00 | 12.56 | 0.60 | 1.83 | 0.23 | 0.77 | 0.16 | **0.15 ± 0.004** |
| R35 | 0.00 | 12.63 | 0.62 | 2.13 | 0.26 | 0.84 | 0.21 | **0.19 ± 0.002** |
| R40 | 0.00 | 12.65 | 0.63 | 1.05 | 0.29 | 0.86 | 0.25 | **0.24 ± 0.001** |
| R45 | 0.00 | 12.67 | 0.63 | 1.07 | **0.30** | 0.98 | 0.32 | 0.31 ± 0.002 |
| R50 | 0.00 | 12.72 | 0.64 | 1.12 | **0.29** | 1.01 | 0.36 | 0.36 ± 0.001 |

* A=3/18 and T=1/8 are different versions of FLUTE and REST, which we illustrate in Appendix B.2.

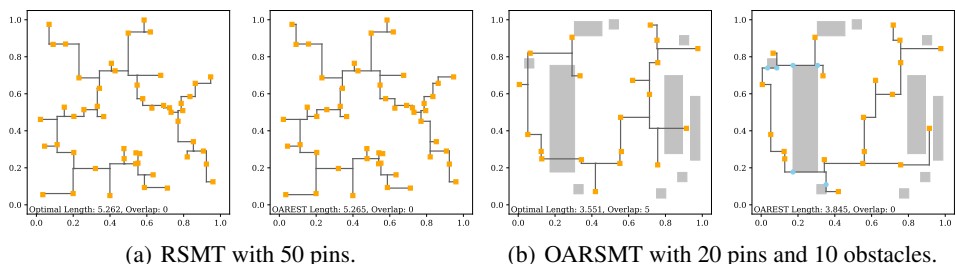

(a) RSMT with 50 pins.   (b) OARSMT with 20 pins and 10 obstacles.

Figure 4: a) A pair of RSMTs with 50 pins, and b) a pair of OARSMTs with 20 pins and 10 obstacles, respectively run by the optimal GeoSteiner (left of each pair) and OAREST (right of each pair).

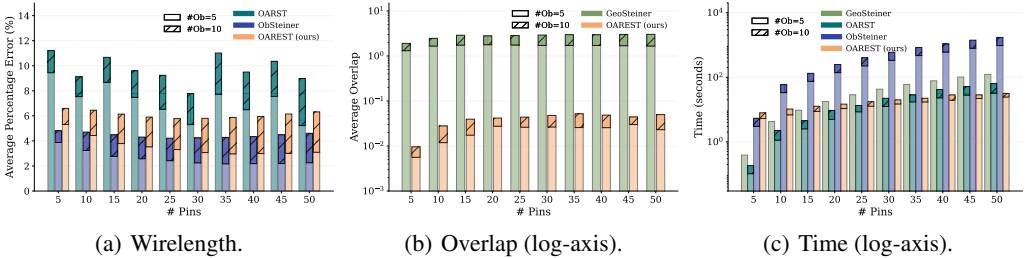

(a) Wirelength.  (b) Overlap (log-axis).  (c) Time (log-axis).

Figure 5: a) Average percentage error on task OARSMT compared to GeoSteiner; b) Average overlap on task OARSMT of GeoSteiner and OAREST; c) Time overhead on task OARSMT of GeoSteiner, OARST, ObSteiner, and OAREST.

Fig. 5(b) shows the average overlaps produced by GeoSteiner and OAREST. As GeoSteiner does not consider obstacles, it inherently suffers from significant overlaps. OAREST, on the contrary, reduces the overlap to a negligible level. Specifically, OAREST achieves >98%/96% success rates (zero-overlap solutions) for instances with 5/10 obstacles, respectively. Detailed statistics can be achieved in Appendix B.1. Such success rates are significant, as no obstacle is visible during the training stage. The small proportion of failed cases (1-4%) typically have minor overlaps in few rectilinear edges after performing OAREST. These can be resolved using maze routing postprocessing [22, 23] with <1% additional runtime compared to OAREST's inference time, maintaining practical feasibility while preserving computational advantages. We leave further reduction of overlaps as future work.

In Fig. 5(c), we compare the time overhead of the four algorithms. Unsurprisingly, the exact OARSMT solver (ObSteiner) has the highest computational cost, reflecting the inherent complexity of solving OARSMT problems exactly. The approximation solution, OARST, achieves a faster runtime than GeoSteiner. Importantly, we are delighted to see that OAREST outperforms all baselines in most instances, with its runtime being minimally affected by the presence of obstacles. This efficiency is attributed to the GPU parallelization capabilities and the linear scalability of OAREST with respect to obstacle handling.

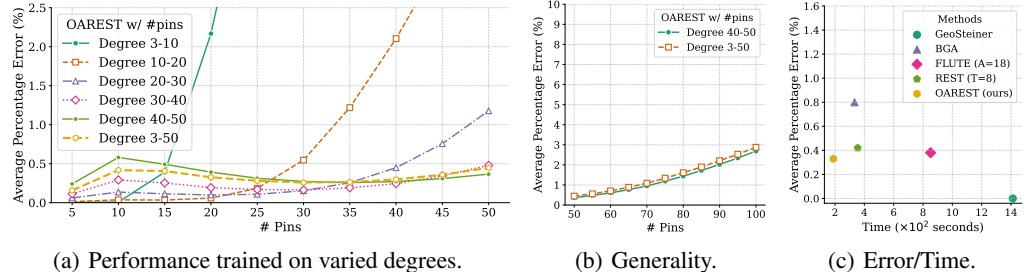

Figure 6: a) Average percentage error of OAREST trained on a varied range of degrees. b) Average percentage error of OAREST $(40 - 50)$ and OAREST $(3 - 50)$ on random instances with degrees $50 - 100$. c) Experiments on ICCAD19 global routing benchmark.

## 4.4 Other Experiments

**Ablation Studies.** Equipped with the dynamic masking strategies, OAREST is capable of training or inferring across a varied range of degrees. Our training strategy first follows REST [5] that trains on a single degree one by one, and is then followed by a quick finetuning process jointly across a range of degrees, including $(3, 10), (10, 20), (20, 30), (30, 40), (40, 50)$, and the full range of $(3, 50)$. As shown in Fig. 6(a), OAREST (3 - 50) has a robust performance on all degrees; however, the highest accuracy is consistently achieved by models trained specifically on smaller, targeted ranges of degrees. For instance, though OAREST (3 - 10) performs poorly on large degrees (15 - 50), it ranks first for degrees 5 and 10. This empirical observation could offer users the flexibility to balance parallel inference and accuracy.

**Generality on Untrained Degrees.** OAREST has a strong capability to extend to untrained degrees. To evaluate this, we tested OAREST (40–50) and OAREST (3–50) on RSMT instances with degrees between 50 and 100. As shown in Fig. 6(b), the average percentage error for both models increases as the number of pins grows. However, even for degrees $51 - 100$, which were not included in the training process, the error remains below $3\%$, highlighting the robustness of OAREST in handling unseen degrees.

**Evaluation on Real-world Circuits.** We evaluated OAREST on the ICCAD19 global routing [15] benchmark. In line with REST [5], we focused on instances with degrees between 3 and 100, as instances with degrees $> 100$ constitute only 0.002% of the dataset (see Table 3). In Fig. 6(c), OAREST outperforms baselines in efficiency with the second-best wirelength (inferior to the exact algorithm GeoSteiner). The time of OAREST is nearly half that of REST (T=8) and 1/7 of GeoSteiner, showing the scalability on real-world benchmarks.

## 5 Related Work

### 5.1 Rectilinear Steiner Minimum Tree (RSMT)

Rectilinear Steiner Minimum Tree (RSMT) algorithms can be mainly divided into approximation, heuristics, and machine learning (ML). Early approximation approaches target near-optimal solutions with efficiency, including R-MST [17] providing 1.5-approximation, [24] approaching 1.25-approximation, and Arora [25] providing $(1 + \varepsilon)$-approximation. Apart from these methods, heuristic-based approaches achieve greater efficiency and accuracy, where BGA [18] utilizes heuristics to optimize edges based on the pre-computed R-MSTs. FLUTE [26, 19], a widely used industrial tool, leverages a lookup table for small instances with $\leq 9$ pins and partitions larger instances into smaller sets for processing. GeoSteiner [16] is also heuristic-based but achieves exact solutions by generating all candidate Steiner Trees.

ML-based methods, especially reinforcement learning (RL), are used for constructing Steiner trees in RSMT. REST [5] proposes the rectilinear edge sequence (RES) and an RL framework based on actor-critic networks to generate the edge sequence, and EPST [6] extends the problem to Octilinear Steiner Minimum Tree (OSMT). Moreover, HubRouter [2] applies the RL-based RSMT to global

routing problems. NN-Steiner [27] utilizes four neural network components to replace the consuming dynamic programming in [25].

Apart from the ability to handle obstacles, our OAREST addresses the multi-degree GPU parallelization in current RL-based methods, which is crucial in VLSI designs.

## 5.2 Obstacle-Avoiding Rectilinear Steiner Minimum Tree (OARSMT)

Based on the vanilla RSMT problems, various works study the strategies to avoid obstacles. Traditional solutions [22, 23] propose heuristic approaches based on maze routing, while EBOARST [28] utilizes a four-step algorithm to efficiently handle obstacles. FOARS [29] addresses obstacles based on FLUTE [26]. [20] proposes an OARSMT algorithm by sequentially using OASG, and OARST, achieving improved wirelength. Moreover, GSLS [30] proposes a guiding solution-based local search method to solve the OARSMT problem, while [31] focuses on selecting Steiner points to handle obstacles. Apart from the approximation and heuristic algorithms, exact algorithms of OARSMT include [32, 33, 34, 21], which study edge-disjoint full Steiner trees (FSTs) building upon the GeoSteiner [16] in vanilla RSMT.

Similar to RSMT problems, RL is also widely used in OARSMT. For instance, [9] utilizes an RL agent to select Steiner points and follows a Maze-router-based Prim's algorithm to form an OARSMT. [7] trains an RL agent based on the small grids of layouts and predicts the Steiner points. It also depends on a Maze routing as postprocessing. [8] performs a fast RL agent to connect pins, but it requires the pre-construction of Obstacle-Avoiding Spanning Graph (OASG) and separates it into shortest path forest (SPF), and finally, a local optimization strategy is employed for postprocessing.

Though these RL-based OARSMT algorithms can effectively handle obstacles, they either suffer from a large action space [7] or combine with heuristics [8, 9], lacking the efficiency of GPU parallelization.

## 5.3 Beyond RSMT in VLSI design

Beyond RSMT, various works in VLSI design employ reinforcement learning (RL) to address specific problems. For example, MaskPlace [35] uses RL to place macros sequentially during placement. Beyond macro placement, MaskRegulate [36] uses an RL policy to adjust existing layouts with dense rewards. Additionally, HAVE [37] introduces hierarchical adaptive multi-task RL and achieves higher hypervolume. In floorplanning, CBL [38] designs a floorplanner using the Corner Block List representation under an RL framework. Also within an RL framework, FlexPlanner [39] extends the problem to 3D floorplanning. For routing, Liao et al. [40] employ a deep Q-network (DQN) for global routing. Note that RSMT is an essential subproblem in routing, where we envision that OAREST makes a promising contribution.

Beyond VLSI design, combinatorial optimization problems form the fundamental basis of various applications, such as molecule generation [41, 42] and protein-protein docking [43]. Within the domain of EDA, our work provides a train-and-test perspective on generalization, specifically regarding varying degrees and obstacle constraints.

# 6 Conclusion and Outlook

We have introduced an RL-based OARSMT solver that supports multi-degree parallel inference and dynamic obstacle avoidance. The main contributions of this work include the theoretical demonstration of the optimality of the rectilinear edge sequence (RES) in OARSMT problems, as well as empirical validation on both synthetic and real-world benchmarks.

Despite its strengths, OAREST has some limitations that present potential areas for future research: 1) In instances with a large number of pins, the error in OAREST increases due to its training being limited to degrees between 3 and 50. 2) In some extreme scenarios, overlaps may still occur with OAREST. To address these issues, we propose the following directions for future work: 1) Training OAREST on instances with obstacles to further improve its ability to avoid obstacles while maintaining minimal wirelength; 2) Testing OAREST on more complex real-world global routing datasets and incorporating additional constraints, such as overflow; 3) Developing methods to completely eliminate overlaps in OARSMT problems.

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

# A Theoretical Results

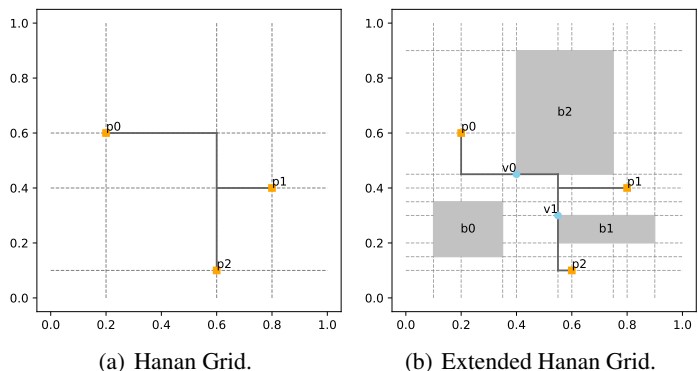

(a) Hanan Grid.  (b) Extended Hanan Grid.

Figure 7: Visualization of a) Hanan Grid, and b) Extended Hanan Grid. The grids are shown as grey dashed lines.

## A.1 Additional Definitions

In this section, we propose additional definitions apart from those in Sec. 2.1. The visualizations of Hanan Grid and Extended Hanan Grid are shown in Fig. 7(a) and Fig. 7(b).

**Definition A.1. [Hanan Grid [44]])** For a given set of pins $\mathcal{P} = \{p_0, p_1, \cdots, p_{n-1}\}$, its Hanan grid $H(\mathcal{P})$ is defined as the grid composed of the intersection points of the horizontal and vertical lines of all pins. Formally,

$$H(\mathcal{P}) = \{(x,y)|x = x_i \text{ or } y = y_i, \forall p_i = (x_i, y_i)\} \tag{9}$$

**Definition A.2. [Extended Hanan Grid]** For a given set of pins $\mathcal{P} = \{p_0, p_1, \cdots, p_{n-1}\}$ and a set of rectangular obstacles $\mathcal{O} = \{o_0, o_1, \cdots, o_{m-1}\}$, its extended Hanan grid $H'(\mathcal{P}, \mathcal{O})$ is defined as the grid composed of the intersection points of the horizontal and vertical lines of all pins and the corners of all obstacles. Formally,

$$H'(\mathcal{P}, \mathcal{O}) = \{(x,y)|x = x_i \text{ or } y = y_i, \forall (x_i, y_i) \in \mathcal{P} \cup \mathcal{V}(\mathcal{O})\}, \tag{10}$$

where $\mathcal{V}(\mathcal{O})$ represents all corner vertices of obstacles $\mathcal{O}$.

## A.2 Assumption and Lemma

In order to prove our main theorem in Sec. 2.3, we first put forward and prove some theoretical findings in this section. Though some theorems/lemmas originate from other works [44, 5], they lack corresponding proofs.

**Assumption A.3.** For a given set of pins $\mathcal{P} = \{p_0, p_1, \cdots, p_{n-1}\}$, its rectilinear edge sequence (RES) is valid iff. such RES connect all pins in $\mathcal{P}$.

**Lemma A.4.** *[Validity of RES] For a given set of pins $\mathcal{P} = \{p_0, p_1, \cdots, p_{n-1}\}$, its rectilinear edge sequence (RES) $res = ((v_0, h_0), (v_1, h_1), \cdots, (v_{n-2}, h_{n-2}))$ is guaranteed to be valid if*

1. *$v_0 \neq h_0$.*

2. *For each rectilinear edge pair $(v_i, h_i)$ $(i \geq 1)$, exactly one of $v_i, h_i$ is visited before by previous rectilinear edge pairs, and the other is not.*

*Proof.* The lemma obviously holds when $n = 3$. For $n > 3$, consider the construction process of RES. Since $v_0 \neq h_0$, the first pair contains two distinct points that are connected. For the $i$-th $(i \geq 1)$ pair, due to the second requirement, the newly-visited pin is connected to the previously-visited pins, and the number of connecting pins plus 1, and so forth, when $i = n - 2$, $n$ pins are connected. $\square$

**Lemma A.5.** *[Hanan's Theorem] For any set of pins $\mathcal{P} = \{p_0, p_1, \cdots, p_{n-1}\}$, there always exists an optimal Rectilinear Steiner Tree (RST), i.e., Rectilinear Steiner Minimum Tree (RSMT), that all Steiner points lie on the Hanan Grid.*

*Proof.* Suppose there exists an optimal RST that has a series of Steiner points $\mathcal{S} = \{s_0, s_1, \cdots, s_{j-1}\}$ not lying on the Hanan Grid. For each $s_i \in \mathcal{S}$, denote its coordinate as $(x_i, y_i)$, then at least one of $x_i, y_i$ is not equal to any of the pin coordinates.

Without loss of generality, we suppose $x_i$ is not equal to any pin coordinates, then we move such Steiner point $s_i$ to the nearest x-coordinate of pins. Due to the characteristic of Manhattan distance, after this movement, the length of the connection path referred to $s_i$ is equal to the previous one.

Repeat the above process on all other Steiner points in $\mathcal{S}$, finally we obtain an optimal RST whose Steiner points are all on the Hanan Grid. $\qquad\square$

**Theorem A.6.** *[Optimality of RES] For any set of pins $\mathcal{P} = \{p_0, p_1, \cdots, p_{n-1}\}$, an optimal RES of $\mathcal{P}$ can always be found such that its corresponding tree is an optimal RSMT for $\mathcal{P}$.*

*Proof.* Let $T^*$ be an optimal RSMT for $\mathcal{P}$ with length $L^*$, and let $\mathcal{R}(\mathcal{P})$ be the set of all valid RES for $\mathcal{P}$. Denote by

$$r^* = \arg \min_{r \in \mathcal{R}(\mathcal{P})} \{\text{length}(T(r))\} \tag{11}$$

an optimal RES in $\mathcal{R}(\mathcal{P})$, where $T(r)$ is the corresponding tree of RES $r \in \mathcal{R}(\mathcal{P})$.

Suppose, for contradiction, that $\text{length}(T(r^*)) > L^*$.

By Lemma A.5, we may assume all Steiner points in $T^*$ lie on the Hanan grid $H(\mathcal{P})$. Consider traversing $T^*$ in a manner similar to a breadth-first search (BFS), starting from an arbitrary pin. Each time a new pin $p_j$ is encountered, it is connected to an already visited pin $p_i$ via a path that follows the Hanan grid and possibly passes through Steiner points.

Since all Steiner points are on the Hanan grid, the path from $p_i$ to $p_j$ consists of horizontal and vertical segments whose total length equals the Manhattan distance between $p_i$ and $p_j$. Therefore, each such path can be represented as a direct sequence of horizontal and vertical edges between pins in the RES framework, implicitly accounting for any Steiner points.

Construct the RES $r'$ by recording each connection $(i, j)$ where $p_j$ is connected to $p_i$. By construction:

1. Each edge $(i, j)$ in $r'$ corresponds to a rectilinear path in $T^*$ with length equal to the Manhattan distance between $p_i$ and $p_j$.

2. The sequence $r'$ satisfies the conditions of Lemma A.4, ensuring that $r'$ is a valid RES.

3. The total length of $T(r')$ equals $L^*$ since each RES edge represents the minimal rectilinear distance required to connect $p_j$ to the existing tree.

Thus, we have $r' \in \mathcal{R}(\mathcal{P})$ and $\text{length}(T(r')) = L^*$. This contradicts our assumption that $\text{length}(T(r^*)) > L^*$. Therefore, it must be that

$$\text{length}(T(r^*)) = L^*, \tag{12}$$

and $r^*$ corresponds to an optimal RSMT. $\qquad\square$

**Lemma A.7.** *[Extended Hanan's Theorem] For any set of pins $\mathcal{P} = \{p_0, p_1, \cdots, p_{n-1}\}$ and a set of rectangular obstacles $\mathcal{O} = \{o_0, o_1, \cdots, o_{m-1}\}$, there always exists an RSMT, that all Steiner points are lying on the extended Hanan Grid $H'(\mathcal{P}, \mathcal{O})$.*

*Proof.* The proof is similar to Lemma A.5. Suppose there exists an optimal RST that has a series of Steiner points $\mathcal{S} = \{s_0, s_1, \cdots, s_{j-1}\}$ not lying on Hanan Grid $H(\mathcal{P})$. For each $s_i \in \mathcal{S}$, denote its coordinate as $(x_i, y_i)$, then at least one of $x_i, y_i$ is not equal to any of the pin coordinates.

Without loss of generality, we suppose $x_i$ is not equal to any of the coordinates of pins, then we try to move such Steiner point $s_i$ to the nearest x-coordinate of pins. Consider two situations:

1. If the movement is successful, then the length of the connection path referred to $s_i$ is equal to the previous one due to the characteristic of Manhattan distance.

2. If the movement fails (caused by obstacles), we can move $s_i$ to the nearest x-coordinate of pins as far as possible until it is adjacent to an obstacle $o_k$. This operation will also not lead to longer length and the Steiner point $s_i$ is moved to the extended Hanan Grid $H(\mathcal{P}, \mathcal{O})$.

Repeat the above process on all other Steiner points in $\mathcal{S}$, finally we obtain an optimal RST whose Steiner points are all on the extended Hanan Grid $H(\mathcal{P}, \mathcal{O})$.

**Assumption A.8.** For a given set of pins $\mathcal{P} = \{p_0, p_1, \cdots, p_{n-1}\}$ and a set of rectangular obstacles $\mathcal{O} = \{o_0, o_1, \cdots, o_{m-1}\}$, its rectilinear edge sequence (RES) $r$ is valid iff.

1. Such RES connect all pins in $\mathcal{P}$.

2. $\forall edge \in r, \forall o_i \in \mathcal{O} : edge \cap \text{int}(o_i) = \emptyset$, where $\text{int}(o_i)$ denotes the interior region of obstacle $o_i$.

$\square$

**Lemma A.9.** *[Existence of valid RES in OARSMT]* *For any set of pins $\mathcal{P} = \{p_0, p_1, \cdots, p_{n-1}\}$ and a set of rectangular obstacles $\mathcal{O} = \{o_0, o_1, \cdots, o_{m-1}\}$, suppose there exists a rectilinear Steiner tree (RST), then there always exists a subset of obstacles $\mathcal{V}'(\mathcal{O})$ and the corresponding valid rectilinear edge sequence (RES) of $\mathcal{P} \cup \mathcal{V}'(\mathcal{O})$.*

*Proof.* Clearly, it holds for the case without obstacles, so we only consider the case when RES overlaps with obstacles.

Define an empty set $\mathcal{U}$ and a RES $r$ of $\mathcal{P}$. Suppose a rectilinear edge $(v_i, h_i)$ in $r$ overlaps with obstacle $o_j \in \mathcal{O}$, then we can choose an unvisited corner vertex $u_j$ of obstacle $o_j$ and divide $(v_i, h_i)$ into $(v_i, u_j), (u_j, h_i)$.

Add $u_j$ to $\mathcal{U}$ and replace $(v_i, h_i)$ with two rectilinear edges $(v_i, u_j), (u_j, h_i)$ in $r$. Traverse all rectilinear edges until none of them overlap with any obstacle. Note that $\mathcal{U} \subset \mathcal{V}(\mathcal{O})$, so we denote $\mathcal{U} \triangleq \mathcal{V}'(\mathcal{O})$.

The split of $(v_i, h_i)$ will simultaneously increase the number of rectilinear edges by 1 and the number of unvisited points by 1, so it satisfies the validity of RES stated in Lemma A.4.

$\square$

## A.3 Proof of the Main Theorem

**Theorem A.10.** *[Optimality of RES in OARSMT]* *For any set of pins $\mathcal{P} = \{p_0, p_1, \cdots, p_{n-1}\}$ and a set of rectangular obstacles $\mathcal{O} = \{o_0, o_1, \cdots, o_{m-1}\}$, an optimal RES of $\mathcal{P} \cup \mathcal{V}'(\mathcal{O})$ can always be found such that its corresponding tree is an optimal OARSMT for $\mathcal{P}$ under obstacles $\mathcal{O}$. Here, $\mathcal{V}'(\mathcal{O}) \subset \mathcal{V}(\mathcal{O})$ is a subset of the corner vertices of all obstacles.*

*Proof.* The proof is similar to that of Theorem A.6. Let $T^*$ be an optimal OARSMT for $\mathcal{P}$ under obstacles $\mathcal{O}$ with length $L^*$ and no overlaps, and let $\mathcal{R}(\mathcal{P}')$ be the set of all non-overlapping and valid RES for $\mathcal{P}' \triangleq \mathcal{P} \cup \mathcal{V}'(\mathcal{O})$. According to Lemma A.9, $\mathcal{R}(\mathcal{P}') \neq \emptyset$. Denote by

$$r^* = \arg\min_{r \in \mathcal{R}(\mathcal{P}')} \{\text{length}(T(r))\} \tag{13}$$

an optimal non-overlapping RES in $\mathcal{R}(\mathcal{P}')$, where $T(r)$ is the corresponding tree of RES $r \in \mathcal{R}(\mathcal{P}')$.

Suppose, for contradiction, that $\text{length}(T(r^*)) > L^*$.

By Lemma A.7, we may assume all Steiner points in $T^*$ lie on the extended Hanan grid $H'(\mathcal{P}, \mathcal{O})$. Consider traversing $T^*$ in a manner similar to a breadth-first search (BFS), starting from an arbitrary pin. Each time a new pin/vertex $p_j$ is encountered, it is connected to an already visited pin/vertex $p_i$ via a path that follows the extended Hanan grid and possibly passes through Steiner points.

Since all Steiner points are on the extended Hanan grid, the path from $p_i$ to $p_j$ consists of horizontal and vertical segments whose total length equals the Manhattan distance between $p_i$ and $p_j$. Therefore,

each such path can be represented as a direct sequence of horizontal and vertical edges between pins in the RES framework, implicitly accounting for any Steiner points.

Construct the RES $r'$ by recording each connection $(i, j)$ where $p_j$ is connected to $p_i$. By construction:

1. Each edge $(i, j)$ in $r'$ corresponds to a rectilinear path in $T^*$ with length equal to the Manhattan distance between $p_i$ and $p_j$.

2. The sequence $r'$ satisfies the conditions of Lemma A.4, ensuring that $r'$ is a valid RES.

3. The total length of $T(r')$ equals $L^*$ since each RES edge represents the minimal rectilinear distance required to connect $p_j$ to the existing tree.

Thus, we have $r' \in \mathcal{R}(\mathcal{P}')$ and $\text{length}(T(r')) = L^*$. This contradicts our assumption that $\text{length}(T(r^*)) > L^*$. Therefore, it must be that

$$\text{length}(T(r^*)) = L^*, \tag{14}$$

and $r^*$ corresponds to an optimal OARSMT. $\qquad\square$

## B  Experiments

### B.1  Success Rate

Table 5: Success rate (%) of GeoSteiner and OAREST on R5-R50 instances with 5/10 obstacles.

| Instances | GeoSteiner | | OAREST (ours) | |
|---|---|---|---|---|
| | 5 Obstacles | 10 Obstacles | 5 Obstacles | 10 Obstacles |
| R5 | 34.45 | 19.68 | 99.51 | 99.11 |
| R10 | 27.65 | 13.60 | 98.55 | 97.64 |
| R15 | 26.57 | 12.78 | 98.41 | 97.02 |
| R20 | 28.03 | 12.03 | 98.06 | 96.38 |
| R25 | 28.15 | 12.48 | 98.01 | 96.26 |
| R30 | 29.75 | 11.37 | 98.14 | 96.44 |
| R35 | 30.21 | 12.70 | 98.35 | 96.36 |
| R40 | 30.37 | 12.10 | 98.35 | 96.52 |
| R45 | 31.50 | 11.86 | 98.27 | 96.69 |
| R50 | 32.29 | 11.98 | 98.57 | 96.71 |

We conduct success rates of GeoSteiner [16] and OAREST in Table 5. Instances with no overlaps are regarded as successes. The failure cases are due to: 1) Generalization error: For example, the R50 data with 10 obstacles occupy 90 points (50 pins + 40 obstacle corners) for each instance, exceeding the maximum number of 50 pins during training. This results in generalization error and could be improved by covering the 50-90 pins during training. 2) Dense obstacles between pins: For some extreme cases when obstacles are dense and the generated rectilinear edges frequently overlap with these obstacles, OAREST will stop updating the activation masking to keep the linearity of batch inference. It is common for one hard sample in a batch to harm the efficiency of the entire batch in machine learning. In our setting, we pay more attention to GPU parallelization for multiple instances.

In some cases, the RL framework can also generate redundant edges in the RES; these are removed in a post-processing step.

### B.2  Baselines

**GeoSteiner [16].** GeoSteiner is an efficient exact algorithm to solve RSMT problems. Based on the characteristics that an optimal RSMT can always be found by combining full Steiner trees only, it first enumerates all possible full Steiner trees and then forms an RSMT. However, the time complexity is exponential to the number of pins.

**R-MST [17].** Rectilinear Minimum Spanning Tree (R-MST) is an approximation approach for RSMT. It can efficiently construct RSMT within $O(n \log n)$ time complexity and is proved [45] to have the length at most 1.5x that of RSMT.

**BGA [18].** BGA employs heuristics to optimize the result. By computing an R-MST first, BGA then repeatedly replace bad edges with better ones to minimize the total length of the R-MST.

**FLUTE [19].** FLUTE computes a look-up table in advance for the instances ≤9 pins, and thus it is an exact algorithm for these instances. For large instances with >9 pins, FLUTE breaks the net into small nets that can be handled by the look-up table. Within this algorithm, the authors introduce an accuracy parameter $A$, which allows users to control the trade-off between accuracy and runtime when generating RSMTs. $A = 3$ is the default accuracy level of FLUTE, where the runtime and error are moderate. Conversely, $A = 18$ represents a higher accuracy level, which invests more computational resources to further reduce the error in wirelength estimation.

**REST [5].** REST is an RL-based framework that trains an actor-critic network on random data. The actor network is responsible for predicting the next rectilinear edge pair to connect while the critic network is utilized to rapidly predict the total wirelength of the tree. The generated rectilinear edge sequence (RES) finally forms an RSMT. In this method, the authors introduce 8 transformations ($T = 8$) that rotate the point set by 0, 90, 180, 270 degrees, with/without the x- and y- axes swapping, and select the best result. These transformations will not change the RSMT solution but bring promising improvement in wirelength. The time overhead of REST ($T = 8$) is approximately $8\times$ that of REST ($T = 1$).

**OARST [20].** OARST is an OARSMT algorithm, but it contains a sequential process, including generating the obstacle-avoiding spanning graph (OASG), the obstacle-avoiding spanning tree (OAST), the obstacle-avoiding rectilinear spanning tree (OARST), and finally obtains the obstacle-avoiding rectilinear Steiner minimal tree (OARSMT). We use its intermediate result of OARST as a strong baseline.

**ObSteiner [21].** ObSteiner implements OARSMT by employing a geometric approach that decomposes the problem into constructing and concatenating full Steiner trees (FSTs) among complex obstacles, while enhancing computational efficiency through virtual terminal additions and pruning strategies.

### B.3   Model Structure

The main structure is revised from [5], which contains an actor network and a critic network. Both of these networks take a set of points $\mathcal{P} = \{p_0, p_1, \cdots, p_{n-1}\}$ as the input and pass through an embedder and an encoder. The actor network has an additional decoder mechanism. During training, $\mathcal{P}$ only contains pins. For inference, $\mathcal{P}$ is a combination of pins and obstacle vertices. Detailed illustration is as follows:

**Embedder.** Given the set $\mathcal{P}$, denote its position matrix as $\mathbf{P} \in \mathbb{R}^{n \times 2}$. Before passing it into the encoder, they are processed by an embedder module to transform the raw input into a higher-dimensional space suitable for encoding. The embeddings are masked by $\mathbf{m}^{\text{input}}$[3]. Specifically, the embedder is defined as:

$$\text{Embedder}(\mathbf{P}) = \text{BatchNorm}(\text{Conv1D}(\mathbf{P}^\top))^\top \circ \mathbf{m}^{\text{input}}, \tag{15}$$

where Conv1D is a $1 \times 1$ convolution that projects the input from 2 to $d_{\text{emb}}$ dimensions, and BatchNorm [46] applies batch normalization to stabilize the embeddings.

The embedder ensures that the input points are transformed into embeddings $\mathbf{E}_{\text{embed}} \in \mathbb{R}^{n \times d_{\text{emb}}}$ with a richer feature representation, which are then fed into the encoder.

**Encoder.** The encoder is a mapping $\text{Encoder} : \mathbb{R}^{n \times d_{\text{emb}}} \to \mathbb{R}^{n \times d_{\text{enc}}}$. Specifically, the Encoder is constructed using a multi-head attention mechanism [47]. The encoder operates with a stack of layers, where each layer is composed of two sublayers: a multi-head attention layer and a position-wise feedforward layer. These sublayers are connected through residual connections, followed by batch normalization to stabilize and accelerate the training process. Formally, given the input $\mathbf{X}$ to an encoder layer:

$$\begin{aligned} \mathbf{X}' &= \text{BatchNorm}(\text{MultiHeadAttention}(\mathbf{X}, \mathbf{X}, \mathbf{X}) + \mathbf{X}), \\ \mathbf{Y} &= \text{BatchNorm}(\text{FeedForward}(\mathbf{X}') + \mathbf{X}'). \end{aligned} \tag{16}$$

---

[3]Note that the vectors in this section are instance-level for simplicity, different from the batch-level matrices in Sec. 3.2.

The multi-head attention mechanism computes attention as:

$$\text{Attention}(\mathbf{Q}, \mathbf{K}, \mathbf{V}) = \text{softmax}\left(\frac{\mathbf{Q}\mathbf{K}^\top}{\sqrt{d_k}}\right)\mathbf{V}, \tag{17}$$

where $\mathbf{Q}, \mathbf{K}, \mathbf{V}$ are the query, key, and value matrices derived from the input $\mathbf{X}$, and $d_k$ is the dimensionality of the keys. Multiple attention heads are concatenated and linearly projected back to the original dimension.

The position-wise feedforward layer applies two linear transformations with a ReLU activation in between:

$$\text{FeedForward}(\mathbf{X}) = \text{ReLU}(\mathbf{X}\mathbf{W}_1 + \mathbf{b}_1)\mathbf{W}_2 + \mathbf{b}_2. \tag{18}$$

This encoder ensures that the representations are enriched with global contextual information while maintaining computational efficiency.

**Forward Pass of Actor Network.** The overall forward pass through the model begins with the embedder. The embeddings are then processed by the encoder, followed by a series of 1D convolutions to extract specific features:

$$\begin{aligned}
\mathbf{E}_{\text{embed}} &= \text{Embedder}(\mathbf{P}), \quad \mathbf{E}_{\text{enc}} = \text{Encoder}(\mathbf{E}_{\text{embed}}), \\
\mathbf{E}_{\text{r}} &= \text{Conv1D}_r(\mathbf{E}_{\text{enc}}^\top)^\top, \quad \mathbf{E}_{\text{x}} = \text{Conv1D}_x(\mathbf{E}_{\text{enc}}^\top)^\top, \quad \mathbf{E}_{\text{y}} = \text{Conv1D}_y(\mathbf{E}_{\text{enc}}^\top)^\top, \\
\mathbf{E}_{\text{xy}} &= [\mathbf{E}_{\text{x}}, \mathbf{E}_{\text{y}}],
\end{aligned} \tag{19}$$

where $\text{Conv1D}_r$, $\text{Conv1D}_x$, and $\text{Conv1D}_y$ are $1 \times 1$ convolution layers applied to the encoder output $\mathbf{E}_{\text{enc}}$, and $[\cdot, \cdot]$ represents concatenation along the feature dimension. This structured processing pipeline ensures that the model captures local and global information from the input points, preparing the features for subsequent decoding tasks.

**Forward Pass of Critic Network.** The critic network is designed to evaluate the quality of a solution by predicting the expected output length, providing a baseline for reinforcement learning. The forward pass of the critic network is as follows:

$$\begin{aligned}
\mathbf{E}_{\text{embed}} &= \text{CritEmbedder}(\mathbf{P}), \quad \mathbf{E}_{\text{enc}} = \text{CritEncoder}(\mathbf{E}_{\text{embed}}), \\
\mathbf{G} &= \text{Glimpse}(\mathbf{E}_{\text{enc}}) = \text{softmax}(\tanh(\mathbf{E}_{\text{enc}})\mathbf{g}')^\top \mathbf{E}_{\text{enc}}, \\
\mathbf{o} &= \text{MLP}(\mathbf{G}),
\end{aligned} \tag{20}$$

where CritEmbedder and CritEncoder respectively have the same structures of Embedder and Encoder with different learnable parameters. Glimpse computes a weighted sum of the encoded representations with $\mathbf{g}' \in \mathbb{R}^{d_{\text{enc}}}$. Finally, a multi-layer perception (MLP) is used that outputs the final predictions $\mathbf{o}$.

**Decoder of Actor Network.** The decoder of the actor network operates as a sequential decision-making process to generate rectilinear edges along with their associated probabilities. We show the whole process in Alg. 1, where the steps marked in blue means that they are executed only for inference phase. This decoder leverages pointer networks [48] with various masking strategies to compute logits. These logits are generated using encoded features and dynamically updated query vectors as inputs. At each step, the decoder selects indices based on computed logits, ensuring constraints such as unvisited points and obstacle avoidance are met. Additionally, the decoder employs activation masks to handle overlaps with obstacles, iteratively refining selections until a valid edge is identified. This mechanism helps the output rectilinear edge sequence and the cumulative log-probabilities adhere to the given spatial and logical constraints.

### B.4 Training Strategy

Our training strategy consists of two phases. The first phase follows the methodology described in [5], while the second phase involves a quick multi-degree finetuning.

In the first phase, the model is sequentially trained from degree 3 to 50. After completing training at degree $t$, the parameters are used to initialize training at degree $t + 1$. Each degree-specific training involves 40,000 iterations with a batch size $B$ that decreases as the degree increases. Specifically, $B$ starts at 4096 for degree 3 and is progressively reduced to 2048, 1024, and 512 for degrees 10, 20, and

---

**Algorithm 1** Actor Decoder

---

**Input:** Encoded features $\mathbf{E}_{\text{enc}}$, $\mathbf{E}_{\text{r}}$, $\mathbf{E}_{\text{xy}}$, obstacles $\mathcal{O}$.
**Output:** RES $r$ and cumulative log-probabilities $p$.
 1: Initialize input mask $\mathbf{m}^{\text{input}}$, visited mask $\mathbf{m}^{\text{visited}}$, obstacle mask $\mathbf{m}^{\text{ob}}$, and activation masks $\mathbf{m}^{\text{act}}$.

 2: Initialize the log-probabilities $p = 0$ of selecting a RES.
 3: Define zero-initialized query vector $\mathbf{q}_0$.
 4: Compute logits $\mathbf{s}_0 = \text{PointerStart}(\mathbf{E}_{\text{r}}, \mathbf{q}_0) \circ (\mathbf{m}^{\text{input}} \wedge \mathbf{m}^{\text{ob}})$.        Select the start point.
 5: Sample starting index $v_0$ from $\mathbf{s}_0$. Denote $v^{(2)} = v_0$.
 6: Update visited mask $\mathbf{m}^{\text{visited}}$.
 7: **for** $j = 1$ to $n - 1$ **do**
 8:     Compute the first query $\mathbf{q}_1$ based on $v^{(2)}$.
 9:     Compute logits $\mathbf{s}^{(1)} = \text{Pointer1}(\mathbf{E}_{\text{r}}, \mathbf{q}_1) \circ (\mathbf{m}^{\text{input}} \wedge \mathbf{m}^{\text{visited}})$.      Select the unvisited point.
10:     Sample or determine first index $v^{(1)}$ from $\mathbf{s}^{(1)}$.
11:     Compute the second query $\mathbf{q}_2$ based on $v^{(1)}$.
12:     Compute logits $\mathbf{s}^{(2)} = \text{Pointer2}(\mathbf{E}_{\text{xy}}, \mathbf{q}_2) \circ (\mathbf{1} - \mathbf{m}^{\text{visited}})$.      Select the visited point.
13:     Sample or determine second index $v^{(2)}$ from $\mathbf{s}^{(2)}$.
14:     Decode indices $x, y$ from $v^{(1)}, v^{(2)}$.
15:     **while** $(x, y)$ overlaps obstacles $\mathcal{O}$ **do**
16:         Update activation masks $\mathbf{m}^{\text{act}}$;
17:         Compute logits $\mathbf{s}^{(1)} = \text{Pointer1}(\mathbf{E}_{\text{r}}, \mathbf{q}_1) \circ (\mathbf{m}^{\text{input}} \wedge \mathbf{m}^{\text{visited}} \wedge \mathbf{m}^{\text{act}})$.      Select the new unvisited point.
18:         Sample or determine first index $v^{(1)}$ from $\mathbf{s}^{(1)}$.
19:         Compute logits $\mathbf{s}^{(2)} = \text{Pointer2}(\mathbf{E}_{\text{xy}}, \mathbf{q}_2) \circ (\mathbf{1} - \mathbf{m}^{\text{visited}})$.      Select the new visited point.
20:         Sample or determine second index $v^{(2)}$ from $\mathbf{s}^{(2)}$.
21:         Decode indices $x, y$ from $v^{(1)}, v^{(2)}$.
22:     **end while**
23:     Add the rectilinear edge $(x, y)$ to RES $r$.
24:     Compute $\log p(v^{(1)}), \log p(v^{(2)})$ from $\mathbf{s}^{(1)}, \mathbf{s}^{(2)}$ and add to $p$.
25:     Update visited mask $\mathbf{m}^{\text{visited}}$.
26: **end for**

---

40, respectively. The Adam optimizer [49] is employed with an initial learning rate of $2.5 \times 10^{-4}$, which decays by a factor of 0.96 after each degree's training.

In the second phase, we leverage a dynamic masking strategy to jointly train the model across a range of degrees, from $n_1$ to $n_2$. The Adam optimizer is again utilized, this time with a learning rate of $5 \times 10^{-5}$. The total number of iterations in this phase is limited to the number of iterations required for training a single degree in the first phase. Specifically, we conduct training for the degree ranges $(n_1, n_2)$ as follows: $(3, 10), (10, 20), (20, 30), (30, 40), (40, 50)$, and $(3, 50)$.

### B.5    Inference Strategy

For the RSMT problems, we use OAREST (3-10) to test R5 and R10, OAREST (10-20) to test R15 and R20, and so forth. For each group of instances, a batch size of $100\,\text{k}/degree$ is used for parallel inference. To further enhance the performance, we use 8 transformations proposed by [5] that rotate the point set by 0, 90, 180, 270 degrees, with/without the x- and y- axises swapping, and select the best result. These transformations will not change the RSMT solution but bring promising improvement in wirelength.

For the OARSMT problems, we use OAREST (40-50) for all groups of instances due to large occupations of obstacles. To demonstrate the efficiency of GPUs, we use the full batch size, i.e., $10\,\text{k}$, for parallel inference. To keep the linearity of inference, the model only inspects the obstacles once for the 'while' in Alg. 1. Here, 8 transformations are also used.

## B.6   Visualization

In this section, we present the visualized results of OARSMTs with 5-50 pins and 0/5/10 obstacles. When no obstacles are present, the OARSMT results are the same as the vanilla RSMTs. As shown in Fig. 8-17, each line represents three pairs of instances with the same number of pins, including the pair without obstacles (left), with 5 obstacles (middle), and with 10 obstacles (right). Within each pair, the left means the result obtained by the exact RSMT algorithm GeoSteiner [16] and the right represents the result obtained by OAREST.

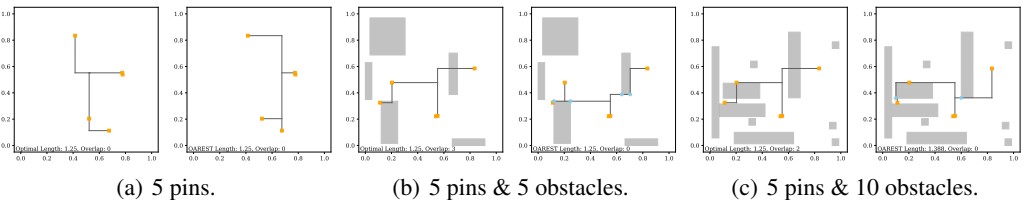

(a) 5 pins.  (b) 5 pins & 5 obstacles.  (c) 5 pins & 10 obstacles.

Figure 8: OARSMT with 5 pins and 0/5/10 obstacles.

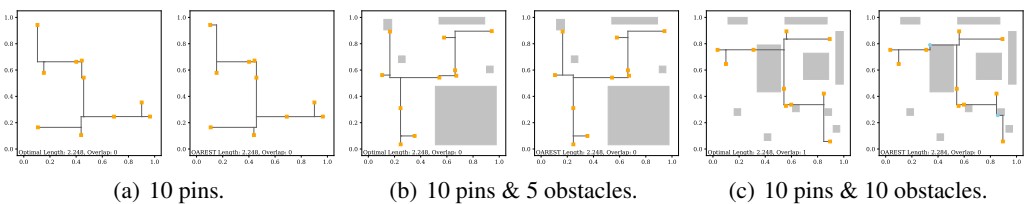

(a) 10 pins.  (b) 10 pins & 5 obstacles.  (c) 10 pins & 10 obstacles.

Figure 9: OARSMT with 10 pins and 0/5/10 obstacles.

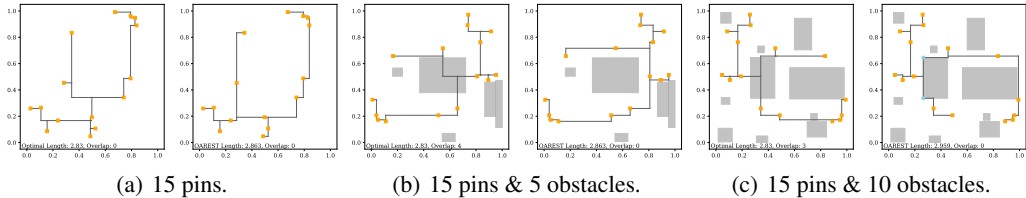

(a) 15 pins.  (b) 15 pins & 5 obstacles.  (c) 15 pins & 10 obstacles.

Figure 10: OARSMT with 15 pins and 0/5/10 obstacles.

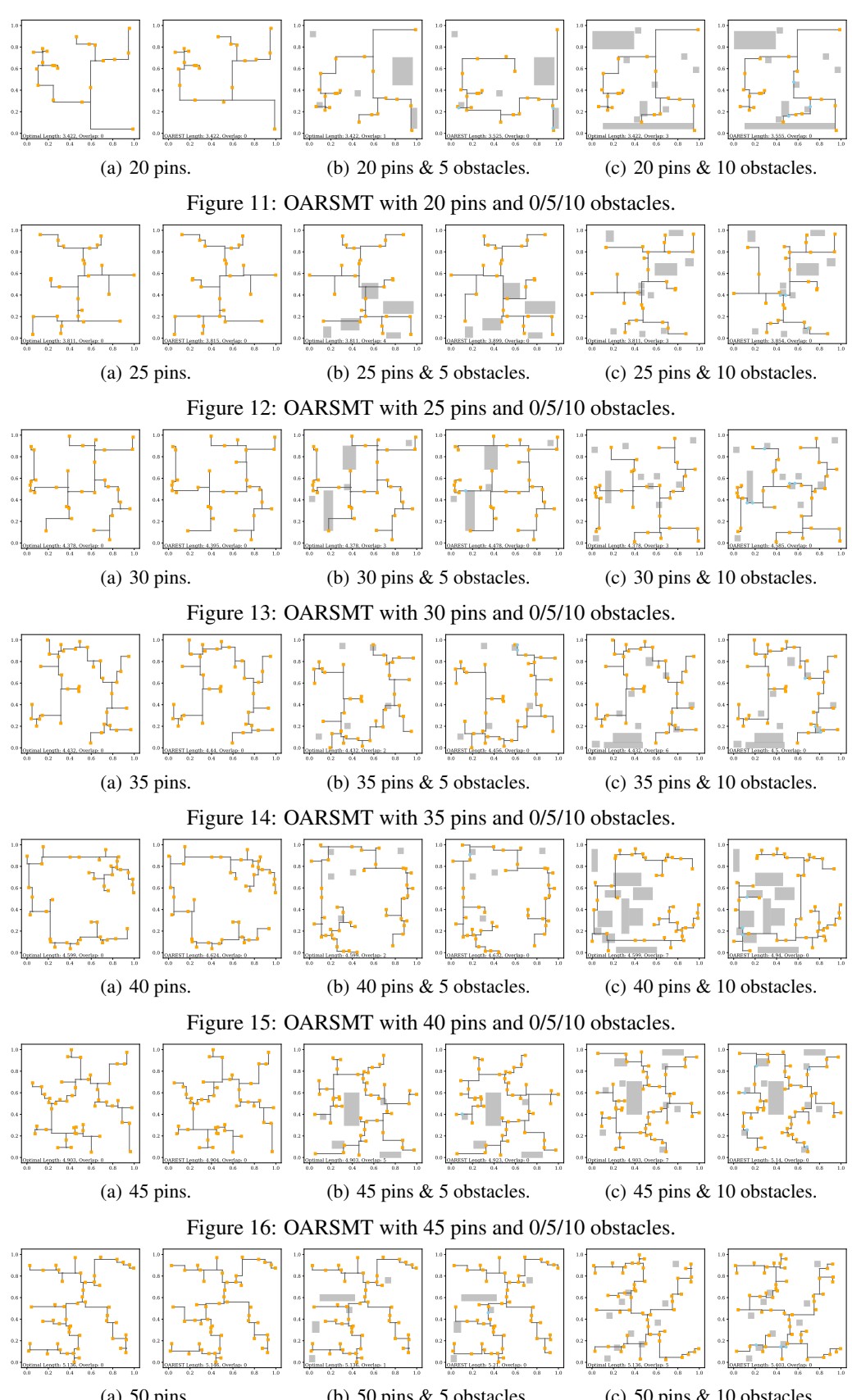

(a) 20 pins. (b) 20 pins & 5 obstacles. (c) 20 pins & 10 obstacles.

Figure 11: OARSMT with 20 pins and 0/5/10 obstacles.

(a) 25 pins. (b) 25 pins & 5 obstacles. (c) 25 pins & 10 obstacles.

Figure 12: OARSMT with 25 pins and 0/5/10 obstacles.

(a) 30 pins. (b) 30 pins & 5 obstacles. (c) 30 pins & 10 obstacles.

Figure 13: OARSMT with 30 pins and 0/5/10 obstacles.

(a) 35 pins. (b) 35 pins & 5 obstacles. (c) 35 pins & 10 obstacles.

Figure 14: OARSMT with 35 pins and 0/5/10 obstacles.

(a) 40 pins. (b) 40 pins & 5 obstacles. (c) 40 pins & 10 obstacles.

Figure 15: OARSMT with 40 pins and 0/5/10 obstacles.

(a) 45 pins. (b) 45 pins & 5 obstacles. (c) 45 pins & 10 obstacles.

Figure 16: OARSMT with 45 pins and 0/5/10 obstacles.

(a) 50 pins. (b) 50 pins & 5 obstacles. (c) 50 pins & 10 obstacles.

Figure 17: OARSMT with 50 pins and 0/5/10 obstacles.

