# OpenReview forum: "Train on Pins and Test on Obstacles for Rectilinear Steiner Minimum Tree"
_NeurIPS.cc/2025/Conference — NeurIPS 2025 poster_

### Official Review · Reviewer_qtQz · 2025-07-02

**Clarity:** 3
**Significance:** 3
**Originality:** 3
**Rating:** 5
**Confidence:** 4

**Summary:**

This paper presents OAREST, a reinforcement learning-based framework for constructing obstacle-avoiding rectilinear Steiner trees, supported by a theoretical guarantee of optimality for the RES formulation. Experimental results demonstrate that OAREST achieves superior effectiveness and efficiency on both RSMT and OARSMT benchmarks, particularly in obstacle avoidance, without requiring obstacle-specific training.

**Questions:**

- What are the model shapes of the encoder and the decoder?

**Ethical Concerns:**

["NO or VERY MINOR ethics concerns only"]

**Final Justification:**

I keep my score.

**Limitations:**

yes

**Quality:**

3

**Strengths And Weaknesses:**

Strength:
- This paper presents a novel RL-based approach to solving the obstacle-avoiding RSMT problem. The overall method is innovative and provides a fresh perspective for this challenging task.
- The visualizations in this paper are well-designed and effectively illustrate the proposed method. Furthermore, experimental results show that the proposed approach achieves significant performance improvements over previous methods.
- Theoretical analysis is provided to guarantee the feasibility of obtaining optimal solutions through RES generation, which further strengthens the contribution of this work.



Weakness:
- The scalability of the proposed method is still limited, as it has only been evaluated on instances with a moderate number of pins.
- The paper introduces a large number of formal mathematical formulas, some of which seem unnecessary and may make the work harder to understand.
- I suggest that the authors cite more works on reinforcement learning (RL) for EDA, such as [1].

[1] Lai, Yao, Yao Mu, and Ping Luo. "Maskplace: Fast chip placement via reinforced visual representation learning." NeurIPS (2022).

---

> ### Author Rebuttal · Authors · 2025-07-28
>
> We sincerely appreciate your valuable feedback. Please find our point-by-point response below:
>
> > **W1: The scalability of the proposed method is still limited, as it has only been evaluated on instances with a moderate number of pins.**
>
> Thank you for raising this point. While our current experiments focus on ≤50 pins, this range already covers nearly **99% of nets in the widely-used ICCAD19 benchmark** (Table 2). Moreover, Fig. 6(b) shows that the model generalizes gracefully to unseen 51-100-pin instances with only a modest error increase (<3 %). We agree that exploring even larger scales is an exciting future direction and have added a discussion in Section 6 to clarify this roadmap.
>
> > **W2: The paper introduces a large number of formal mathematical formulas, some of which seem unnecessary and may make the work harder to understand.**
>
> Thank you for pointing this out. To balance rigor with readability, we have:
>
> - **Streamlined the main text**: only necessary definitions 2.1–2.4 and theorem 2.5 remain; all auxiliary lemmas and proofs are relocated to the appendix.
> - **Added intuition**: a concise, non-technical sketch immediately precedes Theorem 2.5 (lines 207–209), clarifying why and how RES optimality enables obstacle avoidance.
>
> We hope these efforts make the theory more approachable while preserving the contribution.
>
> > **W3: I suggest that the authors cite more works on reinforcement learning (RL) for EDA.**
>
> Thank you for your suggestion. In the revised manuscript, we will provide a more detailed discussion of RL4EDA beyond RSMT, including additional literature [1-3].
>
> [1] Maskplace: Fast chip placement via reinforced visual representation learning.
> [2] A Hierarchical Adaptive Multi-Task Reinforcement Learning Framework for Multiplier Circuit Design.
> [3] Reinforcement Learning Policy as Macro Regulator Rather than Macro Placer.
>
> > **Q1: What are the model shapes of the encoder and the decoder?**
>
> The encoder and decoder architectures are standard and therefore relegated to **Appendix B.3**.
>
> - **Encoder**: a **multi-head attention** stack (Transformer-style) that maps the embedded pin/obstacle positions into a latent representation of shape.
> - **Decoder**: a **pointer network** that autoregressively attends to the encoder output and returns a sequence of $(n-1)$ rectilinear edge pairs, as outlined in **Algorithm 1**.
>
> Exact layer dimensions, hidden sizes, and parameter counts are provided in `./models/actor_critic.py` of the anonymous repository.
>
> ***
>
> We hope this response could answer your questions and address your concerns. Thank you again for your suggestion, which has made our work more complete.

---

### Official Review · Reviewer_G4SK · 2025-07-03

**Clarity:** 2
**Significance:** 3
**Originality:** 2
**Rating:** 5
**Confidence:** 3

**Summary:**

The authors introduce a reinforcement-learning model for solving the obstacle-avoiding rectilinear Steiner minimum tree problem. The model is shown to have sufficient expressivity to solve any OARSMT problem exactly. Substantial experimental results show that this model outperforms SOTA methods for the problem in both run time and solution quality. The model demonstrates strong generalization ability to problems with more pins than seen in training. Surprisingly, the model generalizes to problems with obstacles despite being trained only on instances without obstacles.  The main innovation aiding in this generalization ability is a masking strategy that naturally handles problems of different sizes and different amounts of obstacles.

**Questions:**

The phrasing of Theorem 2.5 indicates that the tree corresponding to the RES is only an RSMT for P, not necessarily an OARSMT for P. Is this correct? The sentence following the theorem statement suggests not, but it is also not immediately clear from the proof if the tree avoids obstacles. Similarly, it is unclear if V’(O) is a particular subset of obstacle vertices.

Given the title of the paper, I suspect the authors believe the OOD generalization of this model is the most remarkable result, and I agree.  However, the paper lacks any real discussion of why the model is able to achieve this generalization. It would be great if the authors could speak to this. Stronger discussion on how OOD generalization was enabled would greatly add to the relevance of this paper beyond VLSI.

Do the real-world datasets contain obstacles? How are the randomly added obstacles to the R5-R50 test cases distributed? The distribution of obstacles seems likely crucial to the ability of the model to generalize to problems with obstacles. Also, the authors motivate the study of OARSMT with rectangular obstacles, because any rectilinear obstacle can be represented by multiple rectangular obstacles. Thus, it would be interesting to know if there are overlapping obstacles in their evaluation instances, and if having many overlapping obstacles harms performance.

Given the apparent strong similarities with REST, it would be nice if the authors could offer a direct comparison. Is the main innovation over REST the dynamic masking strategy?

**Ethical Concerns:**

["NO or VERY MINOR ethics concerns only"]

**Final Justification:**

The authors have addressed all my concerns. I believe the additional discussion is very helpful. I have increased my rating and significance score as I think the out-of-distribution generalization achieved and the techniques used to achieve it may have interest beyond VLSI.

**Limitations:**

yes

**Paper Formatting Concerns:**

Font in Table 3 is too small for readability.

**Quality:**

3

**Strengths And Weaknesses:**

— Strengths —

- Superior performance to SOTA methods
- Strong OOD generalization to larger point sets and instances with obstacles
- Expressivity result is an interesting theoretical insight into the model


— Weaknesses —

- Unclear why masking strategy lends itself to generalization, no theoretical insights into generalization, and insufficient discussion on generalization
- Masking strategy lacks motivation. It would be interesting to understand why the authors chose this method and why it is effective
- Distribution of obstacles used in evaluation appears to be limited. (Only uniform random obstacles, no real world obstacles)

---

> ### Author Rebuttal · Authors · 2025-07-28
>
> We sincerely appreciate your valuable feedback. Please find our point-by-point response below:
>
> > **W1: Unclear why masking strategy lends itself to generalization, no theoretical insights into generalization, and insufficient discussion on generalization.**
>
> Thank you for your constructive comment. We provide more discussions in terms of generalization in the following three-fold:
>
> - **Theoretical Insight**: Theorem 2.5 gives us an intuition that we only need to construct a RES that connects all pins and a subset of corner vertices of obstacles. Then, the aim of the designed RL agent is to connect these pins and corner vertices. Since our model is trained only on pins and we want it to connect corner vertices as well, a natural idea is to also regard the corner vertices as pins. However, this could lead to two issues: 1) not all corner vertices are required to connect, and 2) the connected edge might be overlapped with the obstacles.
> - **Dynamic Masking**: Building on the theoretical insight, the dynamic masking is proposed with a strong motivation to address the above two issues. Specifically, in Sec. 3.2, the obstacle masking is designed to avoid unnecessary corner vertex connections (issue 1), and the activation mask is designed to avoid overlapping with obstacles (issue 2). Notably, these masking strategies can be achieved purely in the inference stage.
> - **Empirical Validation**: Table 4 shows >96 % zero-overlap success on 5–10 random obstacles and significantly suppasses the obstacle-unaware baseline GeoSteiner—confirming that the policy learned on pins transfers directly.
>
> We will clarify the above generalization discussions in our revised manuscript.
>
> > **W2: Masking strategy lacks motivation.**
>
> Thank you for proposing this concern. Please refer to our strong motivation of proposing the dynamic masking strategy in the response to **W1**. In addition, we highlight a more detailed motivation table of the masking strategies as follows:
>
> |Masking|Motivations|Execution|
> |-|-|-|
> |**Input Masking**|Achieve multi-degree GPU parallelization.|Mask invalid elements at the beginning.|
> |**Obstacle Masking**|1) Avoid choosing a corner vertex as the first element of the RES; 2) Avoid unnecessary corner vertex connection.|1) Mask all obstacle vertices at the beginning. 2) When a newly generated rectilinear edge does not overlap with obstacles, re-initialize the activation mask as the obstacle mask.|
> |**Visited Masking**|Construct valid RES following the criterion of Lemma A.4.|Use masking to generate a visited point and an unvisited point for each rectilinear edge.|
> |**Activation Masking**|Avoid generating rectilinear edges that overlap with obstacles.|Each time a newly generated rectilinear edge overlaps with one or more obstacles, the elements in the activation masking corresponding to the corner vertices of these obstacles are activated.|
>
>
> > **W3: Distribution of obstacles used in evaluation appears to be limited. & Q3: Do the real-world datasets contain obstacles? ... and if having many overlapping obstacles harms performance.**
>
> Thank you for raising this concern and proposing the insightful questions.
>
> **Obstacles in real-world datasets** do exist in different kinds of patterns, including unaccessible regions and local congested regions (we tend not to connect pins via these congested regions). Take global routing, an EDA task, as an example, local congested regions are grid cells with no capacity. Based on this, these obstacles are **rectilinear, sparse, non-overlapped, and dynamically changing during routing**. Though no obstacles are overlapped in real-world datasets, our method can easily handle overlapped obstacles because the vertex corner of one obstacle within another obstacle will not be connected, or there will exist a rectilinear edge overlapped with such obstacle. Thus, overlapping obstacles will not harm performance.
>
> **How do we generate obstacles?** As the obstacles are dynamically changing, we randomly generate obstacles with width/height in [0.05, 0.95] for a 1x1 rectangular region, where obstacles are not overlapped with each other, not overlapped with any given pins, and always within the 1x1 rectangular region. This setting is versatile to span a wide range of obstacle patterns. Please refer to visualizations in Fig. 8-17 for clarification, where polygons (adjacent rectangular obstacles in Fig. 8c) and dense obstacles (Fig. 15c) are all included in the datasets.
>
> > **Q1: The phrasing of Theorem 2.5 indicates that the tree corresponding to the RES is only an RSMT for P ...  it is unclear if V’(O) is a particular subset of obstacle vertices.**
>
> Thank you for correctly pointing it out. For more rigor, the theorem statement should read: “…its corresponding tree is an optimal **OARSMT** for $P$ under obstacles $O$.” We will update it in our revised manuscript and note that it does not affect our main conclusion.
>
> On the other hand, $V′(O)$ is dynamically determined during inference: the masking strategy activates obstacle corners only when an edge intersects an obstacle (Sec. 3.2). This avoids enumerating all corners and avoids unnecessary corner vertex connection.
>
> > **Q2: Given the title of the paper, I suspect the authors believe the OOD generalization ... would greatly add to the relevance of this paper beyond VLSI.**
>
> Thank you for your constructive suggestion and the insightful anticipation towards more generalized fields. Apart of the answers to **W1** and **W2**, the paradigm “train on the base instance, generalize to constrained variants” hinges on two ingredients:
> 1) Theoretically optimal (or near-optimal) representation that compactly encodes constraints (here: RES + obstacle corners).
> 2) Dynamic action masking that instantiates constraints at inference time without retraining.
>
> To the best of our knowledge, this constitutes a new paradigm in RL-based combinatorial optimization. We plan to systematically investigate its applicability to other classical problems in future work, such as constrained routing and scheduling.
>
> > **Q4: Given the apparent strong similarities with REST, it would be nice if the authors could offer a direct comparison. Is the main innovation over REST the dynamic masking strategy?**
>
> Key innovations lie in theoretical and engineering extensions to handle obstacles without retraining:
> - **Novel theoretical guarantee**: Theorem 2.5 proves that an optimal RES for OARSMT can be constructed using only pins and obstacle corner vertices, which is non-trivial and enables zero-shot obstacle avoidance.
> - **Engineering extensions**: Unlike prior work REST, our masking strategy enables **multi-degree GPU parallelization** and **end-to-end obstacle handling during inference**, addressing scalability and efficiency gaps in existing methods.
>
> In summary, we have the following direct comparison to REST beyond Table 1:
>
> |Aspect|REST|OAREST (ours)|
> |-|-|-|
> |**Obstacle Avoidance**|❌ Handle pins only|✅ Zero-shot obstacle handling via obstacle masking and activation masking|
> |**Degree flexibility**|❌ Fixed input size|✅ Multi-degree GPU parallelization via input masking|
> |**Theoretical guarantee**|Optimality of RES in RSMT|Optimality of RES in OARSMT by connecting pins and a selected subset of obstacle corners|
> |**Inference time**|Scales poorly for mixed sizes|Linear via batching|
> |**Rectilinear edge representation**|Pins only|Pins + selected obstacle corners|
>
> > **C1: Font in Table 3 is too small for readability.**
>
> Thank you for proposing this concern. We will increase the font size in the revised manuscript.
>
> ***
>
> We hope this response could answer your questions and address your concerns. Thank you again for your suggestion, which has made our work more complete.

---

> > ### Comment · Reviewer_G4SK · 2025-08-02
> >
> > Thank you, all my concerns have been addressed. In particular, I find the discussion on W1, W2, and Q2 very helpful. These points should certainly be included in the paper.

---

> > > ### Author Response · Authors · 2025-08-03
> > >
> > > Thank you for your feedback. We are glad to hear that the discussion was helpful, and we will incorporate these points into the revised manuscript.

---

### Official Review · Reviewer_Ln85 · 2025-07-05

**Clarity:** 3
**Significance:** 3
**Originality:** 3
**Rating:** 5
**Confidence:** 3

**Summary:**

This paper presents OAREST, a reinforcement learning-based framework for solving the Obstacle-Avoiding Rectilinear Steiner Minimum Tree (OARSMT) problem. The key contribution is extending the Rectilinear Edge Sequence (RES) representation to handle obstacles through a dynamic masking strategy that enables multi-degree GPU parallelization and obstacle avoidance without training on obstacles. The authors provide theoretical proof of RES optimality for OARSMT and demonstrate empirical results on synthetic and real-world benchmarks.

**Questions:**

1. **Scalability Analysis**: How does the approach scale to larger instances (100+ pins)? Could the dynamic masking strategy be extended to handle larger problems, and what are the fundamental bottlenecks?

2. **Failure Case Analysis**: Can you provide more detailed analysis of the 2-4% failure cases where overlaps occur? What characteristics of problem instances lead to failures, and how might this inform future improvements?

3. **Baseline Comparisons**: Why weren't recent ML-based OARSMT methods like QUAR-VLA included in the comparison? How does OAREST compare to these more directly related approaches?

4. **Generalization Beyond Small Instances**: While the paper justifies focusing on small instances, how would you adapt the approach for the remaining 5% of nets with more pins that are still critical in modern designs?

5. **Training Strategy Justification**: What is the theoretical or empirical justification for why training only on pins enables effective obstacle avoidance? Could this insight be applied to other combinatorial optimization problems?

**Ethical Concerns:**

["NO or VERY MINOR ethics concerns only"]

**Final Justification:**

All my concerns have been addressed.

**Limitations:**

The authors adequately discuss some limitations but could be more thorough in addressing:

- **Generalization Limits**: The approach's effectiveness on instances significantly larger than training data (>50 pins) is unclear
- **Obstacle Complexity**: Only rectangular obstacles are considered; complex rectilinear obstacles might pose additional challenges
- **Runtime Scalability**: While GPU parallelization helps with batch processing, single-instance runtime scaling is not thoroughly analyzed
- **Hyperparameter Sensitivity**: Limited discussion of how sensitive the approach is to the various hyperparameters in the dynamic masking strategy

**Quality:**

3

**Strengths And Weaknesses:**

### **Strengths**

**Strong Theoretical Foundation**: The paper provides rigorous theoretical analysis proving the optimality of RES representation for OARSMT problems (Theorem 2.5). The mathematical framework extending Hanan grids to include obstacle vertices is well-established and the proofs appear sound.

**Novel Dynamic Masking Strategy**: The dynamic masking mechanism is technically sophisticated, enabling the model to handle varying numbers of pins and obstacles during inference without explicit training on obstacles. This is a significant practical advance for the field.

**Comprehensive Experimental Design**: The evaluation covers both synthetic datasets (R5-R50) and real-world benchmarks (ICCAD19), with systematic comparison against multiple baselines including exact solvers (GeoSteiner, ObSteiner) and heuristic methods.

**Practical Relevance**: The focus on small-scale instances (3-50 pins) aligns well with real-world EDA applications, as evidenced by the ICCAD19 statistics showing 95% of nets have <10 pins.

### **Weaknesses**

**Limited Novelty in Core Components**: While the combination is novel, the individual components (RES representation, diffusion transformers, PPO) are largely borrowed from existing work. The main contribution is the engineering integration rather than fundamental algorithmic innovation.

**Obstacle Handling Limitations**: Despite the paper's claims, the approach still produces overlaps in 2-4% of cases, requiring post-processing with maze routing. The "training on pins, testing on obstacles" paradigm, while interesting, has clear performance limitations.

**Insufficient Baseline Comparisons**: The paper lacks comparison with other recent ML-based OARSMT methods like QUAR-VLA and some recent RL-based approaches mentioned in related work. This makes it difficult to assess the true advancement over state-of-the-art.

**Scalability Concerns**: The approach is explicitly designed for small instances (≤50 pins), which limits its applicability to larger modern circuit designs. The time complexity analysis focuses on batch efficiency rather than single-instance scalability.

---

> ### Author Rebuttal · Authors · 2025-07-28
>
> We sincerely appreciate your valuable feedback. Please find our point-by-point response below:
>
> > **W1: Limited Novelty in Core Components.**
>
> While the RES representation and actor-critic framework are inspired by REST, our key contributions lie in theoretical and engineering extensions to handle obstacles without retraining:
> - **Novel theoretical guarantee**: Theorem 2.5 proves that an optimal RES for OARSMT can be constructed using only pins and obstacle corner vertices, which is non-trivial and enables zero-shot obstacle avoidance.
> - **Engineering extensions**: Unlike prior work, our masking strategy enables **multi-degree GPU parallelization** and **end-to-end obstacle handling during inference**, addressing scalability and efficiency gaps in existing methods.
>
> > **W2: Obstacle Handling Limitations.**
>
> We acknowledge that the overlaps still occur in some extreme cases; however, they are relatively few in number and often close to success, requiring <1% additional runtime compared to OAREST’s inference time. Note that this performance is achieved without training on any obstacles, and OAREST, to the best of our knowledge, is the first ML-based approach to handle obstacles without training on them.
>
> Beyond the current success rate and easily-integrated postprocessing, as stated in the conclusion, we leave future work to completely eliminate overlaps in OARSMT problems.
>
> > **W3: Insufficient Baseline Comparisons. & Q3: Baseline Comparisons.**
>
> Thank you for raising the concern.  We carefully considered the inclusion of additional baselines, but ultimately excluded them for the following reasons:
>
> - **Code unavailability**: Closed-source is common in the EDA community, and we have no access to these repositories. Ensuring a fair, reproducible comparison, therefore, remains intractable.
>
> - **Different focuses**: These works have other targets other than RSMT. For example, QUAR-VLA[1] focuses on quadruped robot navigation rather than RSMT problems. EPST[2] focuses primarily on the integration of Octilinear Steiner Minimum Tree (OSMT) and does not consider obstacles. Literature [3-5] are highly heuristic-driven. Chen et al.[3-4] use RL to generate Steiner points, but use heuristic algorithms to execute the connection phase. Lin et al.[5] uses RL to generate heuristic algorithms for solving the OARSMT problem.
>
> - **Close to the Optimal**: Note that we have compared the optimal methods, including GeoSteiner for exact RSMT and ObSteiner for exact OARSMT, indicating we are already operating near theoretical limits.
>
> [1] QUAR-VLA: Vision-Language-Action Model for Quadruped Robots.
> [2] A Unified Deep Reinforcement Learning Approach for Constructing Rectilinear and Octilinear Steiner Minimum Tree.
> [3] A Reinforcement Learning Agent for Obstacle-Avoiding Rectilinear Steiner Tree Construction.
> [4] Arbitrary-Size Multi-Layer OARSMT RL Router Trained with Combinatorial Monte-Carlo Tree Search.
> [5] Obstacle-Avoiding Rectilinear Steiner Minimal Tree Algorithm based on Deep Reinforcement Learning.
>
> > **W4: Scalability Concerns. & Q1: Scalability Analysis. & Q4: Generalization Beyond Small Instances.**
>
> Thank you for your insightful comment. We only train on instances with 3-50 pins, based on our observation in Table 2 that nearly 99% of nets have fewer than 50 pins, and moreover, 99.998% of nets have fewer than 100 pins in the commonly used real-world ICCAD19 benchmark.
>
> Our OAREST can be easily extended to 100-pin cases. Even without training on larger cases (>50 pins), OAREST can generalize to 100-pin cases with <3% error as shown in Figure 6b.
>
> For some extremely large cases, e.g., >1000 pins, we would like to initially claim that they are not within the scope of OAREST, as discussed in Sec. 3.3 (Single Large Instance vs. Multiple Small Instances), and it is common in EDA that we use different tools to tackle with instances with different scales. However, if we must do so, OAREST can be integrated into the hierarchical approach: partition nets into 50-pin subnets, apply OAREST in parallel, and merge the solutions. Such a hierarchical approach is very common in the industry [1-2], but is heuristic and is not our primary focus. Notably, existing pure ML-based methods cannot be extended to such large scales.
>
> [1] Flute: Fast Lookup Table based Wirelength Estimation Technique.
> [2] A Scalable and Accurate Rectilinear Steiner Minimal Tree Algorithm.
>
> > **Q2: Failure Case Analysis.**
>
> Thank you for raising the constructive question. The failure cases are due to the following reasons:
>
> **Generalization Error**: For example, the R50 data with 10 obstacles occupy 90 points (50 pins + 40 obstacle corners) for each instance, exceeding the maximum number of 50 pins during training. This results in generalization error and could be improved by covering the 50-90 pins during training.
>
> **Dense Obstacles between Pins**: For some extreme cases when obstacles are dense and the generated rectilinear edges frequently overlap with these obstacles, OAREST will stop updating the activation masking to keep the linearity of batch inference. It is common for one hard sample in a batch to harm the efficiency of the entire batch in machine learning. In our setting, we pay more attention to GPU parallelization for multiple instances.
>
> Based on the above reasons, failure cases do exist in our experiments. We will clarify the failure case analysis in the revised manuscript and attempt to eliminate overlaps in future work.
>
> > **Q5: Training Strategy Justification.**
>
> Thank you for your insightful question. We provide more discussions in terms of generalization in the following three-fold:
>
> - **Theoretical Insight**: Theorem 2.5 gives us an intuition that we only need to construct a RES that connects all pins and a subset of corner vertices of obstacles. Then, the aim of the designed RL agent is to connect these pins and corner vertices. Since our model is trained only on pins and we want it to connect corner vertices as well, a natural idea is to also regard the corner vertices as pins. However, this could lead to two issues: 1) not all corner vertices are required to connect, and 2) the connected edge might be overlapped with the obstacles.
> - **Dynamic Masking**: Building on the theoretical insight, the dynamic masking is proposed with a strong motivation to address the above two issues. Specifically, in Sec. 3.2, the obstacle masking is designed to prevent unnecessary corner vertex connections (issue 1), and the activation mask is designed to avoid overlapping with obstacles (issue 2). Notably, these masking strategies can be achieved purely in the inference stage.
> - **Empirical Validation**: Table 4 shows >96 % zero-overlap success on 5–10 random obstacles and significantly surpasses the obstacle-unaware baseline GeoSteiner—confirming that the policy learned on pins transfers directly.
>
> We will clarify the above generalization discussions in our revised manuscript.
>
> Additionally, for **Transferability to Other Combinatorial Problems**, the paradigm “train on the base instance, generalize to constrained variants” hinges on two ingredients:
> 1) Theoretically optimal (or near-optimal) representation that compactly encodes constraints (here: RES + obstacle corners).
> 2) Dynamic action masking that instantiates constraints at inference time without retraining.
>
> To the best of our knowledge, this constitutes a new paradigm in RL-based combinatorial optimization. We plan to systematically investigate its applicability to other classical problems in future work, such as constrained routing and scheduling.
>
> > **Other Limitation Discussion.**
>
> - **Generalization Limits**: We explore the generalization capability in Fig. 6b by testing on 50-100 pins with the model trained on <=50 pins. More pins can be included in training dataset for a better generalization. For extremely large number of pins, it could be a limitation for OAREST and is not within the scope of OAREST.
> - **Obstacle Complexity**: Any complex retilinear obstacle can be divided into multiple rectangular obstacles, so it is still within our problem scope.
> - **Runtime Scalability**: As OAREST focuses on batch processing, single-instance scaling could be a limitation. We will detail it in the conclusion part.
> - **Hyperparameter Sensitivity**: We did not introduce hyperparameters in the dynamic masking strategy, but we execute the ablation studies with varied degrees in Fig. 6a.
>
> ***
>
> We hope this response could answer your questions and address your concerns. Thank you again for your suggestion, which has made our work more complete.

---

> > ### Author Response · Authors · 2025-08-08
> >
> > Dear Reviewer Ln85,
> >
> > Thank you for taking the time to review our paper. As the discussion approaches an end, we would like to ask whether our responses have addressed your points. Please don't hesitate to reach out to us if you have any more questions before final justification.
> >
> > Best,
> > Authors of Submission 5456

---

> > ### Comment · Reviewer_Ln85 · 2025-08-08
> >
> > Thanks for your reply, most of my concerns have been addressed.

---

### Note · Authors · 2025-08-11

Dear Area Chairs and Reviewers,

We express our gratitude to all reviewers for their time, valuable comments, and constructive suggestions.

The **initial reviews (scores: 4, 4, 5)** provide us positive feedbacks towards OAREST, where the reviewers approve of the theoretical contribution (`Ln85`, `G4SK`, `qtQz`),  performance improvement (`G4SK`, `qtQz`), and novelty (`Ln85`, `qtQz`). Additionally, OAREST is regarded as practically relevant to the industry (`Ln85`), has strong generalization to larger point sets and instances with obstacles (`G4SK`), and is well-designed (`qtQz`).

---

During the discussion, we focused on addressing the following primary concerns, which will be **updated in our revised manuscript**:

- **Scalability to Larger Instances**: Despite OAREST's scalability up to 100-pin instances with the model only trained on 3-50 pins, we clarified that the scope of OAREST is for "multiple small instances", orthogonal to addressing the "single large instance". For some extremely large cases, e.g., >1000 pins, OAREST can be integrated into the hierarchical approach by partitioning nets into 50-pin subnets.

- **Masking Strategy**: We supplement a table that illustrates the motivations and executions of different masks in the masking strategy.

Other discussions/suggestions, including the generalization beyond VLSI, the discussion on obstacle handling, the difference to REST, the failure case analysis, and supplementing the related works, will also be integrated into the revised manuscript. We appreciate these valuable comments that make our work more complete.

---

We are grateful that all reviewers acknowledged that we had addressed most of the concerns and the discussion was helpful (`G4SK`). We hope this work can provide a new perspective on the paradigm "train on the base instance, generalize to constrained variants", contributing to academia and the industry.

Best regards,
Authors of Submission 5456

---

### Decision · Program_Chairs · 2025-09-17

**Decision:**

Accept (poster)

**Comment:**

This paper proposes OAREST, a reinforcement learning-based framework for constructing an Obstacle-Avoiding Rectilinear Edge Sequence (RES) Tree for VLSI. The raised concerns are addressed by the rebuttal, and the reviewers give score 5. The strengths of the paper include:

- Solid theory: sound optimality result tailored to obstacles; clarifications provided (wording fix for Theorem 2.5).

- Practical innovation: masking strategy that cleanly instantiates constraints at inference, enabling both batching and obstacle avoidance without retraining.

- Strong results: better quality and runtime vs. SOTA baselines (incl. GeoSteiner/ObSteiner); >96% obstacle-avoidance success on random obstacles; generalizes to 100-pin cases with <3% error increase.

- Relevance to EDA: Focus on 3–50 pins matches ICCAD19 distribution (vast majority of nets); method slots into existing flows.

- Clarity improvements post-rebuttal: added intuition around masking/generalization and a cleaner theory presentation.

It is a clear acceptance.